# Understanding the Emergence of Seemingly Useless Features in Next-Token Predictors

**Mark Rofin**[1]*, **Jalal Naghiyev**[2,3], **Michael Hahn**[4]

[1] School of Computer and Communication Sciences, EPFL
[2] Technical University of Munich, [3] Zuse School ELIZA
[4] Saarland University Campus, Saarland University

## Abstract

Trained Transformers have been shown to compute abstract features that appear redundant for predicting the immediate next token. We identify which components of the gradient signal from the next-token prediction objective give rise to this phenomenon, and we propose a method to estimate the influence of those components on the emergence of specific features. After validating our approach on toy tasks, we use it to interpret the origins of the world model in OthelloGPT and syntactic features in a small language model. Finally, we apply our framework to a pretrained LLM, showing that features with extremely high or low influence on future tokens tend to be related to formal reasoning domains such as code. Overall, our work takes a step toward understanding hidden features of Transformers through the lens of their development during training.

**Website:** https://markfryazino.github.io/useless-features-iclr

**Code:** https://github.com/Markfryazino/useless-features-iclr-code

## 1 Introduction

Large Language Models (LLMs) are usually pretrained with the objective of next-token prediction (NTP). In this paradigm, a model learns to predict each token in a sequence given all previous tokens: in other words, it learns the distribution $p(x_{t+1} \mid x_1 \cdots x_t)$.

Thus, the model is incentivized to compute features that help predict the immediate next token. Hence, one could reasonably expect that the hidden representations at position $t$, computed by a model trained in this way, would contain only the information relevant for predicting $x_{t+1}$. On certain synthetic tasks, this was found to be true, highlighting a downside of NTP as a training objective (Bachmann & Nagarajan, 2024; Thankaraj et al., 2025).

However, a growing body of work on LLMs (and NTP-trained Transformers in general) shows that sometimes they learn much more than that. For example, Transformers reconstruct abstract features of the input text (Templeton et al., 2024; Park et al., 2024), infer the high-level structure of the processes generating their training data, forming 'world models' (Li et al., 2023; Karvonen, 2024; Shai et al., 2024; Jin & Rinard, 2024; Gurnee & Tegmark, 2023), or implicitly predict the sequence multiple tokens ahead (Pal et al., 2023; Jenner et al., 2024). Motivated by these intriguing findings, we ask:

*How do Transformers trained for NTP learn features that don't help in the prediction of the immediate next token?*

The prior work investigating learned features in Transformers mostly employed a teleological perspective: that is, features are viewed in the context of their role in the algorithms implemented by a trained model (e.g., Ameisen et al. (2025); Arditi et al. (2024)). This approach is useful to find the circuits encoded in LLMs, but it doesn't tell us much about the gradient signal that causes those circuits to develop during training. Thus, the ways of how training for NTP drives the emergence of features has been largely underexplored so far.

---

*Correspondence to: mark.rofin@epfl.ch

Towards closing this gap, we develop a novel view on features learned by Transformers. Based on the structure of information flow in causally masked Transformers, we show that features can in principle be learned by three distinctive mechanisms, which we refer to as *direct learning*, *pre-caching*, and *circuit sharing*. The two latter ones allow the token distribution at positions $> i + 1$ to influence the model's representations at position $i$, unlocking the learning of "useless" features. Next, for a given feature, we propose an experimental method to classify it depending on which mechanism contributed the most to its development. We then use our framework to understand the learned features in Transformers trained on different data domains, including toy functions, the board game of Othello, and language.

**Our key contributions include:** (i) A theoretically grounded explanation for why Transformers trained for NTP learn complex features that are not immediately helpful; (ii) An approach for tracing the gradient components of the NTP objective that led to the development of a given feature in a model; (iii) Novel findings obtained using the proposed framework, including an explanation of the OthelloGPT world model fragility, inspection of the role of pre-caching in text generation, and interpretation of the pre-cached features in an LLM.

## 2 SETUP

We use $\mathcal{X}$ to denote a variable-length input space of discrete token sequences $x_1 \ldots x_n$. We view a model $T_\theta$ as representing a function $x_1 \ldots x_n \to \hat{x}_2 \ldots \hat{x}_{n+1}$ such that

$$\hat{x}_{i+1}(x) = h_\theta^{L+1}(r_i^L) \qquad r_{\theta,i}^0(x) = h_\theta^0(x_i)$$
$$r_{\theta,i}^k(x) = h_\theta^k(r_1^{k-1} \ldots r_i^{k-1}), \ k > 1$$

and $r_{\theta,i}^k(x) \in \mathbb{R}^d$. Here $h_\theta^0$ and $h_\theta^{L+1}$ are embedding and unembedding layers, respectively, $r_{\theta,i}^k(x)$ are the values of the residual stream, and $h_\theta^k$ are Transformer blocks.

We call a *learned feature* any linear component of the residual stream at a specific layer and position $\langle w_i^k, r_{\theta,i}^k(x) \rangle$, where a vector $w_i^k \in \mathbb{R}^d$ defines the *feature direction*. We informally call a feature of a sequence $x_{<t}$ *NTP-useless* if there exists an optimal next-token predictor for $x_t$ that doesn't compute it. Otherwise, we call that feature *NTP-useful*. For example, NTP-useless features can include the positioning of the board game pieces that don't affect the set of possible next moves, or the surface properties of the sequence that are irrelevant to its continuation. The central question of our work can then be formulated as understanding how NTP-useless features emerge in Transformers.

## 3 INFORMATION FLOW DECOMPOSITION

### 3.1 GRADIENT DECOMPOSITION

We fix position $i$ and layer $k$ and study all information paths in the computational graph of the model, classifying them by how they relate to $r_{\theta,i}^k(x)$. We argue that the gradient training signal can flow to $\theta$ through three types of paths, illustrated in Figure 1.

Firstly, a gradient signal can come from the immediate next-token prediction (*direct learning*). This includes all paths passing through $r_{\theta,i}^k(x)$ and $\hat{x}_{i+1}$ and represents the effect of the information encoded in $r_{\theta,i}^k(x)$ on the prediction of the immediate next token. These paths are colored green in Figure 1. We formalize this by comparing the overall gradient with a gradient after a stop-gradient operator is applied:

$$\nabla_\theta L_{i \text{ (direct)}}^k = \nabla_\theta L_i - \nabla_\theta L_i^{\text{sg}(k,i)}. \tag{1}$$

Secondly, the information encoded in $r_{\theta,i}^k(x)$ affects the loss at positions $j > i$ because attention heads at those positions can attend to position $i$. Thus, a gradient signal can come from prediction loss at future positions, after passing through one or more attention operations (*pre-caching*). This component includes the paths passing through $r_{\theta,i}^k$ and $\hat{x}_j$ for $j > i + 1$ (blue in Figure 1):

$$\nabla_\theta L_{i \text{ (pre-cached)}}^k = \nabla_\theta \sum_{j \neq i} \left[ L_j - L_j^{\text{sg}(k,i)} \right]. \tag{2}$$

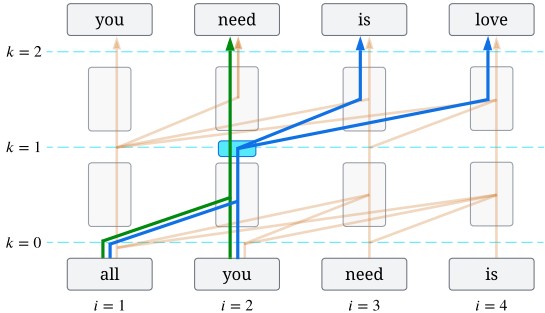 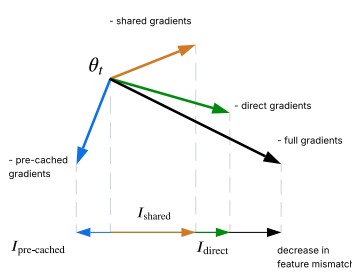

Figure 1: **Left:** An illustration of the information-flow decomposition for $i = 2$ and $k = 1$ into **direct**, **pre-cached**, and **shared** components. **Direct** and **pre-cached** paths must pass through the residual stream at position $i = 2$ and layer $k = 1$; any other path is considered **shared**. The turquoise rectangle indicates $r_{\theta,i=2}^{k=1}$. **Right:** by decomposing the loss gradients, in Section 3.4 we partition the improvements in feature linearity at each training step.

Third, there are paths that do not pass through $r_{\theta,i}^k(x)$ at all. Hence, some gradient signal arises independently of $r_{\theta,i}^k(x)$. Since Transformer blocks use the same parameters to perform the computation at every position, this signal may influence the parameters computing it. We call this phenomenon *circuit sharing* and visualize the related paths in orange in Figure 1:

$$\nabla_\theta L_{i \text{ (shared)}}^k = \sum_j \nabla_\theta L_j^{\text{sg}(k,i)}. \tag{3}$$

The term *pre-caching* is borrowed from Wu et al. (2024), who studied it as a possible explanation for look-ahead in LLMs. We discuss the relation of our work to Wu et al. (2024) in Section 6.

These three components provide an *exhaustive decomposition* of the gradient signal:

**Proposition 3.1** (Loss gradients decomposition). *For any layer $k$ and position $i$,*

$$\nabla_\theta L = \nabla_\theta L_{i \text{ (direct)}}^k + \nabla_\theta L_{i \text{ (pre-cached)}}^k + \nabla_\theta L_{i \text{ (shared)}}^k.$$

By Proposition 3.1, for each $i$ and $k$, the total gradients that are backpropagated to the model parameters after computing the loss on one training batch can be split into three terms distinctive in their nature: direct, pre-cached, and shared components.

**How to study pre-caching and circuit sharing?** In light of Proposition 3.1, a natural question arises: how important each of the introduced components is for learning the task, as well as for representing the latent features picked up by the model. The role of the direct component is in some sense trivial: that is the main source of the gradient signal for predicting the immediate next token, thus we concentrate on analyzing pre-caching and circuit sharing. We approach the analysis by two complementary ways: *intervention* (training a model with one of the components ablated, Section 3.2) and *attribution* (quantifying the influence of each component in a training run, Section 3.4).

### 3.2 ABLATING PRE-CACHING AND CIRCUIT SHARING

**Ablating pre-caching.** Wu et al. (2024) proposed *myopic training* – a way to train an LLM that prevents pre-caching. The only difference between myopic and normal training is that all gradients between the loss at the $i$-th position and the activations at the $j$-th position (where $j \neq i$) are blocked. Since the only path for information to flow between the $i$-th and $j$-th tokens is through attention, it is sufficient to stop gradients after computing the $K$ and $V$ matrices (excluding those for the current token). The purpose of myopic training is to block gradients so that the $i$-th token is not incentivized to compute any feature that is NTP-useless for predicting $x_{i+1}$ but useful for being picked up by an attention head later. This way, pre-cached features do not appear.

**Ablating circuit sharing.** To stop learning features shared across positions, we use a technique that we call $m$-*untied training*. We select an index $m$ and use one set of parameters ($\theta_{\leqslant m}$) for the forward pass on all positions before $m$ and another ($\theta_{>m}$) for positions after $m$. This way, we have

two models that are trained only to predict their respective part of the input, but the second model still attends to the KV-cache of the first one. In the extreme $m$-untied + myopic case, the pre-cached gradients do not flow through attention, so the parameters $\theta_{\leqslant m}$ do not depend on the data after the $m$-th token. In this case, our double model resembles a patient with non-communicating left and right brain hemispheres, as in the split brain experiments in neuroscience (Gazzaniga, 2005).

### 3.3 THE ROLES OF PRE-CACHING AND CIRCUIT SHARING

**Pre-caching increases expressivity.** The power of Transformers comes from complex interactions of attention heads that move information between different tokens. Even constructions as simple as an induction head require at least two layers of attention interacting with each other. Disabling pre-caching prevents the Transformer from deliberately learning such constructions. Indeed, if a feature is NTP-useless at all positions, there is no hope for it to be learned and used by later layers. Thus, many constructions requiring more than one layer of attention (such as the ones described by Liu et al. (2022)) will be impossible to learn.

Note that circuit sharing, by contrast, does not have this property, since even with untied training the gradient signal between tokens still passes through. Compared to myopic training, where an optimal solution cannot be reached due to the lack of training signal, SGD in the case of untied training has the signal needed to find the minimum.

**Circuit sharing enables cross-position feature transfer.** While the strength of pre-caching is increased expressivity, circuit sharing has a different unique property: enabling feature transfer across positions. Imagine a feature that is NTP-useless at position $i$ but NTP-useful at position $j$. Due to its usefulness at $j$, it will be learned, and thanks to circuit sharing, it will also be encoded at $i$. In this way, position $i$ will have access to knowledge learned at a different position.

### 3.4 ESTIMATING THE EFFECT OF GRADIENT COMPONENTS ON FEATURE EMERGENCE

So far, we have argued that there are three path types along which the loss signal, via its gradient, can pass to the model parameters. We now use this decomposition to study the extent to which a feature is produced, over the course of training, by each of the three components of the gradient. To this end, we aim to quantify how much each component of the gradient signal pushes the parameters towards developing a feature.

**Definition 3.2.** We call *a feature mismatch* the value

$$R(x \mid \theta_1, \theta_2, w_i^k) = \frac{1}{2} \left( \langle w_i^k, r_{\theta_1,i}^k(x) \rangle - \langle w_i^k, r_{\theta_2,i}^k(x) \rangle \right)^2$$

The feature mismatch quantifies how much the projections of the residual streams onto the feature $w_i^k$ differ between models parameterized by $\theta_1$ and $\theta_2$.

We now want to quantify the extent to which a single gradient update to an intermediate checkpoint $\theta_t$ narrows the feature mismatch when compared to the final checkpoint $\theta^*$. By separately considering the three components of the gradient signal, we will be able to understand what role each plays in the development of the feature. We formalize this in terms of an *influence* $I(\theta, x, y \mid w_i^k, \theta^*, G)$:

**Definition 3.3.** For a vector $G \in \mathbb{R}^{|\theta|}$, we call *the influence of $G$* the value

$$I_i^k(\theta, x \mid w_i^k, \theta^*, G) = \frac{d}{d\varepsilon} R\left( x \mid \theta + \varepsilon G, \theta^*, w_i^k \right) \Bigg|_{\varepsilon=0}.$$

Applying the decomposition from Proposition 3.1, for each feature we define *direct influence*:

$$I_{\text{direct}}(w_i^k, \theta) = I(\theta, x \mid w_i^k, \theta^*, \nabla_\theta L_{i \text{ (direct)}}^k).$$

The definitions of *pre-cached* and *shared* influences are analogous.

*Remark* 3.4 (informal). Consider a model $T_\theta$, trained for $M$ steps of SGD with a small step size $\eta$. Then

$$R(x \mid \theta_0, \theta^*, w) \approx \eta \cdot \sum_{s \in S} \sum_{t=1}^{M} I(\theta_t, x_t \mid w_i^k, \theta^*, \nabla_\theta L_s),$$

where $S = \{\text{direct}, \text{pre-cached}, \text{shared}\}$.

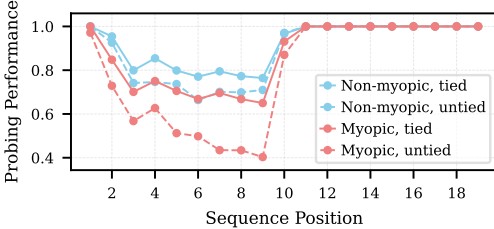 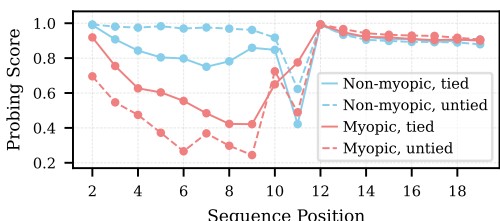

Figure 2: Performance of the linear probes applied to the residual stream after the first Transformer block. **Left:** Majority, the feature "majority-so-far". **Right:** Conditioned Majority, the feature "previous-token". In both cases, the feature is NTP-useless until the 10th token. In both cases, ablating pre-caching and circuit sharing substantially hurts the probing performance.

Remark 3.4 holds approximately due to the first-order approximation of the feature mismatch $R(x \mid \theta_t, \theta^*, w)$. Expressing the change in feature mismatch at each step through its gradient and breaking it down into direct, pre-cached, and shared components leads to the equality above. The remark shows that each loss component has its own influence on feature representation at every step of gradient descent. Integrated over the whole training process, this influence accounts for the discrepancy between the feature representation at the beginning and at the end of training.

The remark above is grounded for SGD; however, the optimizer commonly used to train models is Adam (Kingma & Ba, 2014). Thus, we adjust each gradient component $G$ to reflect the gradient steps made by Adam. We keep separate moments $m_{(\text{component})}$ for each of the three components and calculate the step attributed to each of the components as

$$G_{(\text{component})} = -\alpha \cdot \frac{m_{(\text{component})}}{1 - \beta_1^t}$$

Then the $G_{(\text{direct})}, G_{(\text{pre-cached})}$ and $G_{(\text{shared})}$ are an exact partition of the optimizer step, and we use them to compute influence as defined above.

We interpret the value $\widetilde{I}_{\text{direct}}(w_i^k) \equiv \sum_t I_{\text{direct}}(w_i^k, \theta_t)$ (*integrated direct influence*) as the overall impact of the direct loss component on the emergence of the feature $w_i^k$ in the model, and similarly for the other two components. We use this decomposition to answer which combinations of direct, shared, and pre-cached gradient signals produced a feature over the course of training, by evaluating the magnitudes of the three integrated components $\widetilde{I}_{\text{direct}}(w_i^k), \widetilde{I}_{\text{pre-cached}}(w_i^k)$, and $\widetilde{I}_{\text{shared}}(w_i^k)$.

## 4 EXPERIMENTS WITH SMALL TRANSFORMERS

In all experiments in this section, we train GPT-2-like Transformers (Radford et al., 2019) for NTP using cross-entropy loss. If not stated otherwise, all our models have standard architecture: they are non-myopic and tied. To estimate if a Transformer represents a given feature linearly, we train layer- and position-specific linear probes (Alain & Bengio, 2016; Belinkov, 2022) to predict the value of a feature using the residual stream of the trained model $\theta^*$ as input. For consistency, we always treat probing as a regression task and evaluate the probes using Pearson correlation between the predicted and true values of a feature. Each of the probes represents one feature direction $w_i^k$.

In the attribution experiments, we then retrain the Transformer from scratch with the same random seed and data order, repeating the training trajectory of the first run that lead to $\theta^*$. For each batch in the training set, we compute $I_{\text{direct}}(w_i^k, \theta), I_{\text{pre-cached}}(w_i^k, \theta)$, and $I_{\text{shared}}(w_i^k, \theta)$. We sum those values across the batches, obtaining the integrated influences for each feature[*].

### 4.1 IN TOY TASKS, NTP-USELESS FEATURES DO NOT EMERGE WITHOUT PRE-CACHING AND CIRCUIT SHARING

First, we verify our intuitions using two toy tasks where we have a clear understanding of the required circuits: *Majority* and *Conditioned Majority*. We train two-layer models to solve each task.

In **Majority**, each example $x$ consists of $M$ tokens sampled uniformly from a vocabulary of size $V$, and $M$ tokens sampled from the set of the most frequent tokens so far: $\text{argmax}_t \text{count}(t, x_{\leqslant M})$.

---

[*]Our approach is discussed in more detail in Appendix B.

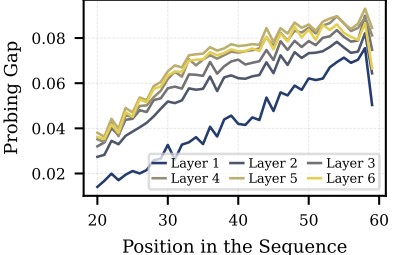

| Component | NTP-useful? | [2.5% | 97.5%] |
|-----------|-------------|-------|--------|
| direct | yes | 2.85 | 12.38 |
| | no | -4.69 | 2.74 |
| pre-cached | yes | -1.99 | 0.66 |
| | no | 0.55 | 3.05 |
| shared | yes | 4.80 | 12.48 |
| | no | 2.93 | 9.91 |
| combined | yes | 12.14 | 19.05 |
| | no | 4.42 | 10.07 |

Figure 3: **Left:** the gap in probing performance between NTP-useful and NTP-useless board squares for OthelloGPT. **Right:** 95% confidence intervals for the integrated influence of each component on the representation of NTP-useful and NTP-useless features.

The task is solved by a simple uniform attention head computing the most frequent token. We track the influence components of the feature $F_1$ "the most frequent token so far."

**Conditioned Majority** is designed to bring out the importance of pre-caching. The input consists again of two parts: $M$ uniformly sampled tokens, followed by $M$ samples from the set of those tokens that followed the token "A" most often in the first part. The task requires a mechanism akin to induction heads (Olsson et al., 2022): the first attention layer attends to the preceding token, and the second layer attends to the tokens after "A." We study the feature $F_2$ "the preceding token is A."

In both tasks, the first $M$ tokens are random and can be predicted trivially, without $F_1$ or $F_2$. Thus, in the input phase $F_1$ and $F_2$ are NTP-useless, and we are interested in whether our models will learn to represent them, and if they do, why.

For different random seeds, we train (non-)myopic ($M$-un)tied models. Thus, we obtain models that all experienced direct learning, but had pre-caching and/or circuit sharing ablated. We plot the performance of probes trained to extract $F_1$ for Majority and $F_2$ for Conditioned Majority in Figure 2. In both cases the myopic untied models show the most fragile representation of the features during the input phase, supporting our prediction that such models have no incentive to learn NTP-useless features. Lifting the ablation of pre-caching or circuit sharing improves the performance of probes. Notably, both tied and untied myopic models are unable to learn Conditioned Majority (Appendix E.1): the two-layer circuit described above cannot develop without pre-caching.

## 4.2 PRE-CACHING AND CIRCUIT SHARING HOLD TOGETHER OTHELLOGPT'S WORLD MODEL

Next, we apply our framework to study Transformers trained to predict legal moves in the game of Othello, a common testbed for evaluating world models implicit in language models (Li et al., 2023; Yuan & Søgaard, 2025). In Othello, two players place their tiles on the cells of an $8 \times 8$ board in turns. The set of legal moves is a deterministic function of the current board state. The initial work on the topic claimed the discovery of coherent board state features in OthelloGPT (Li et al., 2023; Nanda et al., 2023), however more recent evidence was more pessimistic, reporting that the implicit world model of OthelloGPT is fragile, especially when it comes to boards sharing the same set of legal next moves (Vafa et al., 2024; 2025). We aim to make sense of this new evidence from our perspective of NTP-use(ful/less) features.

Recall that we defined NTP-useless features as the ones that are not needed for predicting the next token. Hence, in the case of Othello, for each sequence of moves $x_{<t}$, NTP-useless features are the representations of cells that don't affect the set of the legal next moves $x_t$. Our theory implies that, all else being equal, NTP-useless features should be represented worse than NTP-useful features since they lack the direct learning component. However, pre-cached and shared components are active even for NTP-useless features, supplying them with some training signal.

We follow a standard experimental setting and train a Transformer on randomly generated game transcripts represented as sequences of up to 60 tokens. The $i$-th token indicates the square where the tile was placed during the $i$-th move in the game. Then we train linear probes to extract the board state from the model's hidden representations.

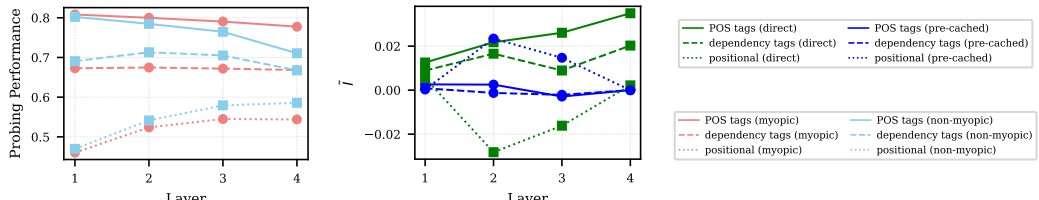

Figure 4: Comparison of the three distinct types of features in a small language model: POS tags, dependency tags, and the positional feature. **Left:** probing scores for myopic and non-myopic models. **Center:** direct and pre-cached influence components for each feature type.

The experimental results align with our predictions. Figure 3 (left) shows the gap between the performance of linear probes trained to extract NTP-useful and NTP-useless features: it is always positive and increases with layer depth (most likely because the deepest layers align more on direct influence). We verify the reasons for this disparity by inspecting the influence of each component calculated separately for NTP-useless and NTP-useful features (Figure 3 (right)). Direct influence alone is positive for NTP-useful but indistinguishable from zero for NTP-useless features. However, since pre-cached and shared influence is positive even for NTP-useless features, they are still learned.

These observations provide new evidence that complements the empirical findings of Vafa et al. (2025). Specifically, we show with statistical significance that the model's inductive bias toward next-token partitions of state arises from the direct gradient component promoting learning of NTP-useful features. We also refine the result of Vafa et al. (2025) by showing that the NTP-useless cells are learned as well, albeit less robustly, since the signal for learning them is present due to the non-zero pre-cached and shared components. Together, our results explain both 1) why the model represents NTP-useful cells better than NTP-useless ones, and 2) why NTP-useless cells can still be recovered better than chance.

### 4.3 Pre-Caching is Required for Coherent Text Generation, but not Needed for Syntax

The environment that ultimately interests us is language. Thus, we train and study tiny Transformers for natural text generation. The main question we ask is: *Is pre-caching needed for coherent text generation, and which relevant features does it affect?*

We use TinyStories v2 (Eldan & Li, 2023), which contains GPT-4-generated children's stories. The texts in this dataset use simple language that could be understood by a child; these properties make it learnable even by a very small model. Following Eldan & Li (2023), we train tiny (non-)myopic GPT-2-like language models on this dataset with different random seeds. For each story, we randomly choose a starting point and from there sample a substring of 64 tokens. We annotate each token with syntactic features (POS and dependency tags), as well as with a positional feature: the position of the token in the original story before subsampling the sequence. See Appendix D.3 for the full experimental details.

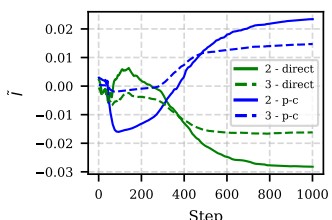

Figure 5: Development of the direct and pre-cached influence components of the positional feature in the non-myopic model during training.

We observe that myopic models have a consistently much higher loss than non-myopic models ($3.29 \pm 0.02$ vs $2.53 \pm 0.10$, with training curves reported in Appendix E.3), indicating that pre-caching is necessary for the task. At the same time, nearly all features we study seem to be direct: the performance of probes extracting these features does not vary significantly between myopic and non-myopic models, and for all but the positional feature, pre-caching influence lies much lower than direct influence (Figure 4) with non-overlapping confidence intervals for mean (Figure 14) and one-sided Wilcoxon tests showing that the gap is statistically significant (Table 5). We conclude that simple syntax can be learned without pre-caching.

What is pre-caching needed for then, if not syntax? It seems to be relevant for computing more complex properties of the input text: as can be seen in Figure 4 (center), in a non-myopic model the

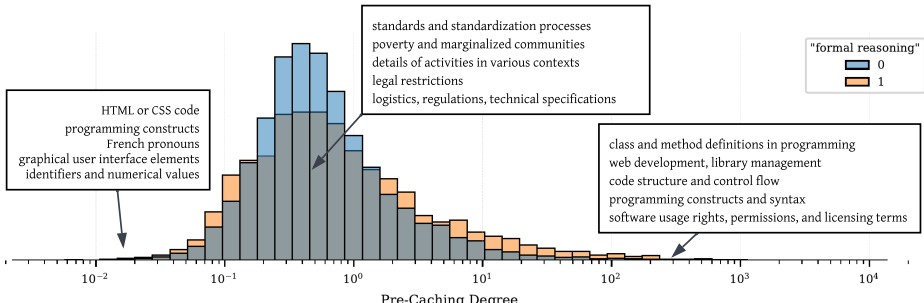

Figure 6: Distribution of $Q(w)$ for the SAE features of Gemma 2 2B. The text boxes show the compressed descriptions of 5 features from the tails of the distribution, as well as 5 random features from the mode of the distribution.

positional feature, which relates to high-level properties of the stories, starts to be learned due to the pre-caching influence. Figure 5 shows a transition during training, when pre-caching starts affecting the feature, hinting to a development of a circuit involving it.

## 5 INVESTIGATING FEATURES IN LARGE LANGUAGE MODELS

Retraining an LLM from scratch is computationally very expensive, which is why the field of LLM interpretability operates mostly on final model checkpoints. For the same reason, the attribution method that we used in Section 4 is inapplicable in the same form to large-scale models.

However, even with access only to the final checkpoints, we can draw some conclusions about the emergence reasons of features, connecting traditional interpretability and our framework introduced above. In this section, we use this connection to understand the latent features of a State-of-the-Art LLM: Gemma 2 (Gemma Team et al., 2024).

### 5.1 FINDING AND UNDERSTANDING PRE-CACHED FEATURES IN AN LLM

The standard way of estimating the causal role of a feature in a Transformer is *an intervention* (Mueller et al., 2024): modifying the activations of a model during the forward pass to alter the representation of a feature and observing the changes in predictions. Assume that one intervenes on the residual stream $r_{\theta^*,i}^k(x)$ of a trained model $T_{\theta^*}$, adding to it a vector $w$, and then records the KL-divergence between the predictions with and without the intervention at each position after $i$:

$$d_j^{/i} = D_{\text{KL}}(T_{\theta^*}(x_j \mid x_{<j}) \| T_{\theta/i}(x_j \mid x_{<j}))$$

Here, $T_{\theta/i}(x_j \mid x_{<j})$ are the predictions of a model under the intervention on the $i$-th position.

**Proposition 5.1** (Approximating influence with interventions).

$$\frac{I_i^k(\theta^{/i}, x \mid w, \theta^*, \nabla_\theta L_{\text{pre-cached}})}{I_i^k(\theta^{/i}, x \mid w, \theta^*, \nabla_\theta L_{\text{direct}})} \approx \frac{\sum_{j>i+1} d_j^{/i}}{d_{i+1}^{/i}} \triangleq Q(w)$$

The proof is deferred to Appendix C. Proposition 5.1 implies that even if we cannot rigorously compute the influence components of a feature as we don't have access to the model's training trajectory, we can estimate the ratio of direct and pre-cached influence around the trained model using the quantity we denote $Q(w)$. In other words, we can find out if a given feature is more likely to be direct (an intervention changes only the prediction of the immediate next token) or pre-cached. Importantly, however, we can only make statements about the components ratio in the region around the final model, not along the whole training path.

**Finding pre-cached features in Gemma 2.** To extract the learned features in an unsupervised fashion, we use a Sparse Autoencoder from the Gemma-Scope suite (Lieberum et al., 2024). To compute $Q(w)$ for each feature, we find the tokens where the feature is active and ablate it by setting its activation value to zero, effectively subtracting $\langle r, w \rangle / \|w\|^2 \cdot w$ from the residual stream.

Then we use the model's predictions with and without the ablation to calculate $d_j^{/i}$ and $Q(w)$ as explained above. See the details in Appendix D.4.

Using the fact that automatically generated descriptions of the SAE features are available, we study the descriptions of features with extreme values of $Q(w)$ and observe that most of them are related to programming or formal structure of the input text. Based on this observation, we form a hypothesis: are features related to formal reasoning tasks more likely to have extreme values of $Q(w)$?

To test this hypothesis, we label each feature as 0 or 1 depending on whether it activates on this type of inputs (details in Appendix D.5), and plot $Q(w)$ separately for those groups (Figure 6). Indeed, the tails of the distribution see much higher concentration of these formal features. Estimated 95% CI for the $\sigma_{\text{formal}}$ when modeled with a log-normal distribution is $1.63 \pm 0.03$, for the $\sigma_{\text{not formal}}$ it is $1.23 \pm 0.02$. The results align with the intuitions detailed in Section 3.3: pre-caching seems to be needed in tasks that require emulating formal computational devices (e.g., AST for code parsing).

To further test the robustness of the link between $Q(w)$ and feature semantics, we examine how steering features with different values of $Q(w)$ affects samples from the model. For each feature, we draw unconditional generations from the model while steering that feature by adding its direction vector, scaled by a steering coefficient, to the residual stream at the target layer. We find that steering features with a high pre-caching degree leads to generations containing more code and more punctuation than average (Figure 19). Interestingly, we don't observe the same effect for features on the opposite end of the spectrum. The results thus support the connection between a feature having a high value of $Q(w)$ and its involvement in formal reasoning. For features with low $Q(w)$, this connection, if present, appears weaker and not as easily detectable by steering. Full experimental details are provided in Appendix D.6.

## 5.2 Pre-Caching and Look-ahead Are Separate Phenomena

Wu et al. (2024) initially introduced the notion of pre-caching as a potential explanation for the look-ahead in LLMs (Pal et al., 2023), suggesting that pre-cached features may be the ones that contribute to the ability to predict future tokens the most. We test this hypothesis by investigating if the learned features of Gemma 2 with high $Q(w)$ are also most useful for the look-ahead.

We obtain a future token predictor with look-ahead distance $k$ by training a linear layer mapping $W_{\text{LA}}$ from the residual stream vector at position $i$ to the token at position $i + k + 1$ on a subset of the Pile dataset (Gao et al., 2020):

$$\hat{x}_{i+t+1} = h_\theta^{L+1}(W_{\text{LA}} \cdot r_{\theta,i}^k(x))$$

Then, for each SAE feature, we compute the angle $A(W_{LA}, w, r)$ between the feature direction $w$ and the subspace formed by $r$ main singular components of $W_{\text{LA}}$. If a feature $w$ is useful for predicting future tokens, it is to be expected that $W_{\text{LA}}$ will learn to use it, and $\cos A(W_{LA}, w, r)$ will be non-zero. We report the Spearman correlation between $\cos A(W_{LA}, w, r)$ and $Q(w)$ for each $r$ in Figure 7 for the mapping up to 5 tokens ahead. See the details of the experiment in Appendix D.7.

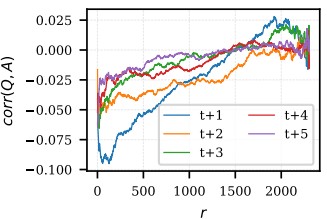

Figure 7: Correlations between $\cos A(W_{LA}, w, r)$ and $Q(w)$.

Surprisingly, we find the negative correlation across all look-ahead maps. This means that not only the look-ahead is not caused exclusively by the pre-cached features only, but they contribute to it less than direct features. This is strong evidence for the breadcrumbs hypothesis of Wu et al. (2024): look-ahead in LLMs arises not from implicit planning but rather from the similarities of features needed to predict tokens at different positions.

## 6 Related Work

**Effects of NTP training.** This paper is a part of a line of work aiming to understand how the NTP objective shapes the algorithms learned by Transformers. Bachmann & Nagarajan (2024); Nagarajan et al. (2025) discuss the critiques of NTP and show how it can lead to learning undesirable shortcuts or lack of creativity, mitigated by multi-token training initially proposed by Gloeckle et al. (2024).

Motivated by these findings, Hu et al. (2025) modify the objective, Thankaraj et al. (2025) propose reordering the tokens in the input.

**World models and future token prediction in Transformers.** Li et al. (2023); Karvonen (2024); Shai et al. (2024); Jin & Rinard (2024), among others, find that Transformers implicitly reconstruct the latent variables of the data generation process, arriving to the so-called "world models". The learned representations, however, are not always linear (Engels et al., 2024) and can be brittle and inconsistent (Vafa et al., 2024; 2025). Pal et al. (2023) show that future token predictions can be decoded from the internal representations of a pretrained LLM with nontrivial accuracy using probes and learned prompts. Jenner et al. (2024) find a similar effect in a neural network trained to play chess.

Most relevant to our work, Wu et al. (2024) investigate the reasons behind the emergence of look-ahead in LLMs and proposes the *pre-caching hypothesis*, stating that some tokens might "prepare in advance" the information relevant for future tokens. We build upon that work and borrow the term "pre-caching" from there. However, our contribution is distinct from that of Wu et al. (2024) in several aspects: most importantly, we bring the level of analysis down to the development of specific features, and also introduce the concept of circuit sharing, not analyzed by Wu et al. (2024).

**Developmental interpretability.** Another relevant to us line of research is the emerging subarea of interpretability that takes inspiration from the field of training dynamics and analyzes the changes in the algorithms encoded in the models at different moments of training, asking when and why LLMs pick up different skills or features during training. Tigges et al. (2024) analyze circuits in LLMs at different training checkpoints and find that circuits implementing various task abilities stay consistent across model sizes and emerge at similar numbers of training steps. Bayazit et al. (2025) track the emergence of features using crosscoders trained across model checkpoints, and Xu et al. (2024) use SAEs for the same purpose. Wang et al. (2025) monitor emerging specialization of attention heads using the concepts from Singular Learning Theory. Kangaslahti et al. (2025) cluster different datapoints based on their loss dynamics, estimating the moments when certain abilities become acquired by the model. Michaud et al. (2023) target explaining the LLM scaling laws by suggesting that scaling the number of parameters and training steps allows them to learn skills of increasing rarity. In a similar vein to our work, Mircea et al. (2025) inspect the gradients during LLMs pretraining and proposes that LLMs may go through a period of "loss deceleration", when the per-example gradients cancel each other, slowing down the learning of new concepts.

# 7 CONCLUSION

Traditional work in mechanistic interpretability aims to understand a learned feature by tracing which circuits it plays a role in. However, those circuits do not simply appear in the model; they need a gradient signal to develop. In this work, we studied the sources of that gradient signal, shifting from the commonly employed static teleological perspective to a developmental one, in which learned features in Transformers are viewed as outcomes of gradient-based learning rather than a gear in a final algorithm.

In Section 4, we showed how this change of perspective makes a difference, helping interpret features that cannot be explained solely through being a part of an algorithm predicting the immediate next token. We believe that Othello is a prime example, where representation of NTP-useless features *can be explained* by inspecting pre-cached and shared gradient components.

Unfortunately, modifying the training pipeline of a model is often too computationally expensive to be done in practice. The best one can do in this case is inspecting the region around the trained model, which is what we did in Section 5, but even using the restricted toolkit of intervening on a static model, we can perform analysis on the development of linearly represented features under study.

Improving the efficiency of our attribution method and adapting it to be applicable to large-scale models is a potential avenue for future work. Another promising direction is to use our method to discover previously unknown interpretable features by analyzing the residual stream subspaces formed by a distinct gradient component (e.g., only by pre-cached updates but not direct updates).

ACKNOWLEDGMENTS

MR thanks Aleksandra Bakalova, Yash Sarrof, Yana Veitsman, Xinting Huang, Michael Rizvi-Martel, Vaishnavh Nagarajan, Naomi Saphra, and Nicolas Flammarion for helpful discussions and feedback on the drafts of this paper. JN is supported by the Konrad Zuse School of Excellence in Learning and Intelligent Systems (ELIZA) through the DAAD programme Konrad Zuse Schools of Excellence in Artificial Intelligence, sponsored by the Federal Ministry of Education and Research.

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

# A  PROOF OF PROPOSITION 3.1

**Proposition A.1** (Restated from Proposition 3.1). *For any layer $k$ and position $i$,*

$$\nabla_\theta L(x, T_\theta(x)) = \nabla_\theta L^k_{i \,(\text{direct})}(x, T_\theta(x)) + \nabla_\theta L^k_{i \,(\text{pre-cached})}(x, T_\theta(x)) + \nabla_\theta L^k_{i \,(\text{shared})}(x, T_\theta(x)) \tag{4}$$

*Where*

$$\nabla_\theta L^k_{i \,(\text{direct})}(x, T_\theta(x)) = \nabla_\theta L(x_{i+1}, T_\theta(x)_i) - \nabla_\theta L\left(x_{i+1}, [T_\theta(x)]^{\text{sg}(i,k)}_i\right), \tag{5}$$

$$\nabla_\theta L^k_{i \,(\text{pre-cached})}(x, T_\theta(x)) = \sum_{j \neq i}\left[\nabla_\theta L(x_{j+1}, T_\theta(x)_j) - \nabla_\theta L\left(x_{j+1}, [T_\theta(x)]^{\text{sg}(i,k)}_j\right)\right], \tag{6}$$

$$\nabla_\theta L^k_{i \,(\text{shared})}(x, T_\theta(x)) = \sum_j \nabla_\theta L\left(x_{j+1}, [T_\theta(x)]^{\text{sg}(i,k)}_j\right) \tag{7}$$

*Proof.*

$$\nabla_\theta L(x, T_\theta(x)) = \nabla_\theta L\left(x, T_\theta(x)\right) + \nabla_\theta L\left(x, [T_\theta(x)]^{\text{sg}(i,k)}\right) - \nabla_\theta L\left(x, [T_\theta(x)]^{\text{sg}(i,k)}\right) =$$

$$= \sum_{j=1}^{N-1} \nabla_\theta L\left(x_{j+1}, [T_\theta(x)]^{\text{sg}(i,k)}_j\right) + \sum_{j=1}^{N-1} \nabla_\theta L\left(x_{j+1}, T_\theta(x)_j\right) - \sum_{j=1}^{N-1} \nabla_\theta L\left(x_{j+1}, [T_\theta(x)]^{\text{sg}(i,k)}_j\right) =$$

$$= \underbrace{\sum_{j=1}^{N-1} \nabla_\theta L\left(x_{j+1}, [T_\theta(x)]^{\text{sg}(i,k)}_j\right)}_{\nabla_\theta L^k_{i \,(\text{shared})}} + \underbrace{\left(\nabla_\theta L\left(x_{i+1}, T_\theta(x)_i\right) - \nabla_\theta L\left(x_{i+1}, [T_\theta(x)]^{\text{sg}(i,k)}_i\right)\right)}_{\nabla_\theta L^k_{i \,(\text{direct})}} +$$

$$+ \underbrace{\sum_{j \neq i}\left[\nabla_\theta L\left(x_{j+1}, T_\theta(x)_j\right) - \nabla_\theta L\left(x_{j+1}, [T_\theta(x)]^{\text{sg}(i,k)}_j\right)\right]}_{\nabla_\theta L^k_{i \,(\text{pre-cached})}}$$

$\square$

# B  ADDITIONAL DETAILS ON ATTRIBUTION EXPERIMENTS

**Computing the component influence.** By Definition 3.3,

$$I^k_i(\theta, x \mid w^k_i, \theta^*, G) = \frac{d}{d\varepsilon} R\left(x \mid \theta + \varepsilon G, \theta^*, w^k_i\right)\bigg|_{\varepsilon=0}$$

A simple application of the chain rule gives an equivalent definition:

$$I^k_i(\theta, x \mid w^k_i, \theta^*, G) = \left\langle \nabla_\theta R\left(x \mid \theta, \theta^*, w^k_i\right), G\right\rangle \tag{8}$$

Thus, computing the influence of a given gradient term can be done by taking the inner product of that gradient term and the gradient of the feature mismatch.

To compute $I_{\text{direct}}(w^k_i, \theta)$, $I_{\text{pre-cached}}(w^k_i, \theta)$, and $I_{\text{shared}}(w^k_i, \theta)$ for given $i$ and $k$, we employ the following algorithm. First, we calculate $\nabla_\theta L_j$ for every $j$ in a standard way. Next, we calculate $\nabla_\theta L^{\text{sg}(k,i)}_j$ in a similar manner, but during the forward pass of the model we detach the tensor corresponding to $r^k_{\theta,i}(x)$. After that, we compute the gradient decomposition terms according to the definitions in Section 3.4.

The only thing left is the gradient of the feature mismatch. We apply the linear probe defined by $w^k_i$ to both $r^k_{\theta,i}(x)$ and $r^k_{\theta^*,i}(x)$ and compute the feature mismatch according to Definition 3.2. We run one more backpropagation to find $\nabla_\theta R$ and take its inner product with the gradient decomposition terms, obtaining the desired influence values.

**Correction for Adam.** As metioned in Section 3.4, the naive computation of influence components described above is grounded for training with SGD (Remark 3.4) but not for Adam. Indeed, in the case of Adam, the parameter update is computed differently, and a linear step toward the negative gradient no longer reflects how the model is trained. Moreover, due to the inclusion of the first and second moments, the parameter update cannot be computed as a linear sum of three separable components:

$$\theta_{t+1} - \theta_t \neq g(\nabla L_{(\text{direct})}) + g(\nabla L_{(\text{pre-cached})}) + g(\nabla L_{(\text{shared})})$$

while for SGD it is true, with $g(x) = -\eta \cdot x$ (and that fact is the basis behind Remark 3.4).

However, with two small adjustments we can make the Adam parameter update separable. First, we keep separate moments for each of the three components:

$$m_t^{(\text{direct})} = \beta_1 m_{t-1}^{(\text{direct})} + (1 - \beta_1)\nabla L_{(\text{direct})}$$

(similarly for the other two).

Second, we use the same adaptive learning rate for all three components:

$$\alpha = \frac{\eta}{\sqrt{\hat{v}_t} + \epsilon},$$

where $\hat{v}_t$ is computed normally. Then,

$$\theta_{t+1} - \theta_t = g(m_t^{(\text{direct})}) + g(m_t^{(\text{pre-cached})}) + g(m_t^{(\text{shared})}),$$

where

$$g(x) = -\alpha \cdot \frac{x}{1 - \beta_1^t}.$$

This way, we can use $g(m_t^{(\text{direct})})$ as an adjusted gradient for the direct component. From here, the computation is similar to the case of SGD.

**Overall Algorithm.** First, for a given dataset, we train a model normally. In all our experiments, we use Adam to follow community standards. After the model has been trained, we freeze it, save the hidden activations on the validation dataset, and obtain a set of linear probes to regress latent variables.

For us, these probes represent the directions $w_i^k$ needed to compute the gradient-component influence. Once we have the probes, we retrain the model from scratch, fully repeating its training trajectory, which involves starting from the same initialization as the first time, keeping the data point order the same, fixing the PyTorch random seed, etc. During this second training run, at each training batch we compute the feature mismatch and the gradient-component influence for each probe.

The second run is needed because during the first one we do not have the directions $w_i^k$ needed to compute component influences, and to estimate them we need $\theta^*$, which itself requires the training run to finish. In principle, one could save all gradient components during the first run and then retrieve them from memory when computing the influence, but that is extremely expensive in terms of the disk space needed. Thus, we perform two consecutive training runs.

## C   PROOF OF PROPOSITION 5.1

**Proposition C.1** (Restated from Proposition 5.1)**.**

$$\frac{I_i^k(\theta^{/i}, x \mid w, \theta^*, \nabla_\theta L_{\text{pre-cached}})}{I_i^k(\theta^{/i}, x \mid w, \theta^*, \nabla_\theta L_{\text{direct}})} \approx \frac{\sum_{j>i+1} d_j^{/i}}{d_{i+1}^{/i}} \triangleq Q(w)$$

*Proof.* Both direct and pre-cached gradient components consider only the gradient paths that have $r_i^k$ as a bottleneck, so by chain rule they can be expressed as

$$\nabla_\theta L_{i\ (\text{direct})}^k = J_\theta[r_i^k] \cdot \nabla_{r_i^k} L_i$$

And

$$\nabla_\theta L^k_{i \text{ (pre-cached)}} = J_\theta[r^k_i] \cdot \nabla_{r^k_i} \left( \sum_{j>i} L_j \right)$$

For convenience, from this point we will treat them together as $\nabla_\theta L_c$, $c \in \{\text{direct}, \text{pre-cached}\}$.

$$R(x \mid \theta, \theta^*, w^k_i) = \frac{1}{2} \left( \langle w^k_i, r^k_{\theta,i}(x) \rangle - \langle w^k_i, r^k_{\theta^*,i}(x) \rangle \right)^2$$

$$\nabla_\theta R(x \mid \theta, \theta^*, w^k_i) = \left( \langle w^k_i, r^k_{\theta,i}(x) \rangle - \langle w^k_i, r^k_{\theta^*,i}(x) \rangle \right) \nabla_\theta \langle w^k_i, r^k_{\theta,i}(x) \rangle = \Delta \cdot J_\theta[r^k_{\theta,i}](w^k_i)$$

Per equation 8,

$$I^k_i(\theta, x \mid w^k_i, \theta^*, \nabla_\theta L_c) = \left\langle \nabla_\theta R\left(x \mid \theta, \theta^*, w^k_i\right), \nabla_\theta L_c \right\rangle = \tag{9}$$

$$= \Delta(w^k_i)^T J_\theta[r^k_{\theta,i}]^T J_\theta[r^k_i] \cdot \nabla_{r^k_{\theta,i}} L_c = J_\theta[r^k_{\theta,i}]^T J_\theta[r^k_{\theta,i}] \langle \Delta w^k_i, \nabla_{r^k_{\theta,i}} L_c \rangle \tag{10}$$

Imagine we do a feature ablation: that is, we take a trained model $\theta^*$ and add a vector $\Delta w^k_i$ to $r^k_{\theta^*,i}$. We measure the loss $L_c$ that depends on the model's output. Doing a first-order approximation around $r_{\theta',i}$ ($\theta'$ are the parameters of this intervened model),

$$L'_c - L^*_c \approx (\Delta w^k_i)^T \nabla_{r_{\theta',i}} L_c \tag{11}$$

A corollary of equation 10 and equation 11 is that for any model with parameters $\theta'$ such that $r^k_{\theta,i} = r^k_{i\theta^*} + \Delta w^k_i$, the influence of component $c$ is proportional to the increase in loss compared to the model $\theta^*$.

Assuming that that $\theta^*$ perfectly fits the data ($p(x_t \mid x_{<t}) = T_{\theta^*}(x_t \mid x_{<t})$, the difference in cross-entropy NTP losses becomes

$$L'_\text{direct} - L^*_\text{direct} = - \underset{x_t \sim T_{\theta^*}(x_t|x_{<t})}{\mathbb{E}} \log T_{\theta'}(x_t \mid x_{<t}) + \underset{x_t \sim T_{\theta^*}(x_t|x_{<t})}{\mathbb{E}} \log T_{\theta^*}(x_t \mid x_{<t}) =$$

$$= \underset{x_t \sim T_{\theta^*}(x_t|x_{<t})}{\mathbb{E}} \frac{\log T_{\theta^*}(x_t \mid x_{<t})}{\log T_{\theta'}(x_t \mid x_{<t})} = D_{\text{KL}}(T_{\theta^*}(x_t \mid x_{<t}) \| T_{\theta'}(x_t \mid x_{<t}))$$

Similarly, for the pre-cached component, the difference in losses corresponds to the sum of KL-divergences for future positions.

$$\frac{I^k_i(\theta, x \mid w^k_i, \theta^*, \nabla_\theta L_\text{direct})}{I^k_i(\theta, x \mid w^k_i, \theta^*, \nabla_\theta L_\text{pre-cached})} = \frac{D_{\text{KL}}(T_{\theta^*}(x_{i+1} \mid x_{\leqslant i}) \| T_{\theta'}(x_{i+1} \mid x_{\leqslant i})}{\sum_{j>i+1} D_{\text{KL}}(T_{\theta^*}(x_{j+1} \mid x_{\leqslant j}) \| T_{\theta'}(x_{j+1} \mid x_{\leqslant j}))} \tag{12}$$

Where $T_{\theta'}$ is a model where $r^k_i$ was modified along the direction $w^k_i$.

$\square$

# D ADDITIONAL EXPERIMENTAL DETAILS

## D.1 TOY TASKS (SECTION 4.1)

Table 1: Hyperparameters for the experiments with the tasks of Majority and Conditioned Majority.

| Hyperparameter | Value | Hyperparameter | Value |
|---|---|---|---|
| Layers | 2 | Steps | 3000 |
| Heads | 4 | Train size | 102,400 |
| Hidden dim | 128 | Eval size | 10,240 |
| Feedforward dim | 512 | Number of seeds | 10 |
| Learning Rate | 0.001 | Input phase size | 10 |
| Batch size | 256 | Output phase size | 10 |
| | | Vocabulary size | 3 |

## D.2 OTHELLO (SECTION 4.2)

Table 2: Hyperparameters for the Othello experiment.

| Hyperparameter | Value | Hyperparameter | Value |
|---|---|---|---|
| Layers | 6 | Steps | 5000 |
| Heads | 4 | Train size | 5,120,000 |
| Hidden dim | 256 | Eval size | 20,480 |
| Feedforward dim | 1024 | Number of seeds | 1 |
| Learning Rate | 0.001 | | |
| Batch size | 1024 | | |

Following prior work, we generate a dataset of synthetic games of Othello. When generating a new game, we always randomly choose a move that is legal given the current board state.

We encode the game as a sequence of tokens, where the $i$-th token represents a square where a stone was placed at the $i$-th turn, irrespective of which player placed it. We also add a token `pass` for the turns when no moves are legal. The games in our datasets have a fixed length of 60 turns, and end with repetitions of `pass` if the game tree finished before that.

When training the linear probes, we encode the board state as a matrix of -1, 0, and 1. 1 represents the squares belonging to the active player, -1 represents the squares belonging to the non-active player, and 0 represents the empty squares. When a square is placed, the board state matrix gets negated (because the active player changed) and -1 gets added to one of the cells.

When estimating the influence separately for NTP-useless and NTP-useful squares, we don't take empty cells into account. For a given board state, we consider a square NTP-useful if flipping it (negating the corresponding cell in the board state matrix) changes the set of the legal moves. Otherwise, the square is considered NTP-useless. When estimating the influence components for the NTP-useless and NTP-useful features, we use the same probe for both types, but compute feature mismatch separately. This way, we aggregate $R(x \mid \theta, \theta^*, w_i^k)$ disjointly for the objects in the batch where the square is NTP-useless and NTP-useful, and proceed as usual.

After that, $\widetilde{I}_{\text{full}}(w_i^k)_{\text{NTP-useless}}$ can be seen as the contribution of NTP-useless features into the development of the linear direction defined by $w_i^k$. If $\widetilde{I}_{\text{full}}(w_i^k)_{\text{NTP-useless}}$ is negative or close to zero, it indicates that these features do not contribute to the linearity of the representation.

Due to the resource constraints, we cannot compute the influence components for each position and square of the board. Thus, we use 16 central squares and compute influence for every 4th position (from 4th to the 56th turn).

## D.3 SMALL LANGUAGE MODEL (SECTION 4.3)

We employ the tokenizer used in the original work on TinyStories (Eldan & Li, 2023).

We test the following features:

1. POS tags: we annotate POS tags using spaCy (Honnibal et al., 2020), one-hot encode them and use the 10 most common tags, encoded as 0 or 1, as features. Since one word can contain multiple tokens, each token gets assigned the tag of the word it belongs to.

Table 3: Hyperparameters for the small LM experiment (TinyStories).

| Hyperparameter | Value | Hyperparameter | Value |
|---|---|---|---|
| Layers | 4 | Steps | 1000 |
| Heads | 4 | Train size | 256,000 |
| Hidden dim | 128 | Eval size | 30,000 |
| Feedforward dim | 512 | Number of seeds | 10 |
| Learning Rate | 0.001 | Max length of text | 64 |
| Batch size | 1024 | | |

2. Dependency tags: same procedure, also the 10 most common tags.

3. Positional feature: since we sample substrings of 64 tokens from the original stories to train the model, the positions of tokens in these substrings do not directly correspond to their positions in the original text. We use that original position as another feature that assumes the values from 0 (the token is the first token in the original story) to 1 (the last token).

Thus, we have 20 binary features and 1 continuous feature. We estimate influence of those features for each position from 30 to 39.

## D.4 CALCULATING $Q(w)$ (SECTION 5.1)

To characterize the causal role of the learned features in Gemma 2 (Gemma Team et al., 2024), we employ a Sparse Autoencoder (SAE) from the Gemma-Scope suite (Lieberum et al., 2024), specifically the SAE trained at the residual stream in layer 15 with 16k hidden features.

The first step is to curate a dataset of diverse, high-activating text examples for each feature. We select text sequences from Neuronpedia (Lin, 2023) where features show their highest activation. For each feature, we find the token position with the maximum activation value and extract the surrounding text, including a fixed window of 10 subsequent tokens to create an initial set of sequences. This set is then filtered for diversity based on token-level Levenshtein distance, resulting in a collection of high-activating and textually different sequences.

On these sequences, we perform an intervention at the token position of maximum activation. We intercept the residual stream activation vector, pass it through the SAE's encoder, and set the activation value of our target feature to zero. This modified set of feature activations is then passed through the SAE's decoder to create a new, ablated residual stream vector, which replaces the original one in the forward pass.

To quantify the impact of the feature, we compare the model's subsequent token predictions with and without the intervention. We measure the change in the output probability distribution at each future position using KL-divergence. This allows us to compute a $Q(w)$ score representing the ratio of the feature's delayed to immediate influence. We average this score across different sequences. As discussed in Appendix C, a high $Q(w)$ value indicates a 'pre-cached' feature whose primary influence is on the model's future state, while a low value indicates a 'direct' feature influencing on the immediate prediction. We filter out from the analysis the features, ablating which does not show substantial effect on any generated tokens:

$$\frac{1}{S}\sum_{s=1}^{S}\sum_{j>i} D_{\mathrm{KL}}(T_{\theta^*}(x_{j+1}^{(s)} \mid x_{\leqslant j}^{(s)}) \| T_{\theta'}(x_{j+1}^{(s)} \mid x_{\leqslant j}^{(s)})) < 0.05$$

After that, we are left with 14,565 features to analyze out of the initial 16,384.

## D.5 CLASSIFYING SAE FEATURES (SECTION 5.1)

To assess whether a given SAE feature relates to formal reasoning, we use its automatically generated decsription available in Neuronpedia (Lin, 2023). We use the available descriptions generated using GPT-4o-mini (OpenAI, 2024) and classify these descriptions themselves with GPT-4.1-nano (OpenAI, 2025).

Our prompt is `Here is the description of a certain textual property:` `"`*DESCRIPTION*`". Is this property related to `*TAG*`? Respond with one word only: yes or no.`

In this prompt, *DESCRIPTION* is filled with the retrieved description of the feature, and *TAG* is one of "`computer code, programming languages, or math`" or "`syntax or text structure`".

In this way, for each feature we obtain the labels `is_code` and `is_syntax` and we consider the feature related to formal reasoning if at least one of them is 1.

### D.6   STEERING SAE FEATURES (SECTION 5.1

To steer a feature $w$ in a model, we add $w$ multiplied by a steering coefficient to the residual stream: $r_i^k \leftarrow r_i^k + \delta w$. In our experiments, we set $\delta = 10$.

For each feature, we sample 64 generations from a model with that feature steered at every token position. We sample 20 tokens unconditionally with temperature 1, thus generating samples from a distribution $T_\theta(x)$, but under steering.

For each generated sample, we count the number of punctuation characters it contains. We also classify whether the sample represents a snippet of computer code using Llama 3.1 8B (Grattafiori et al., 2024) as a classifier. This gives us, for each feature, two metrics: the number of generated code snippets (out of 64 total generations) (`#code`) and the average number of punctuation symbols per generation (`#punct`).

In Figure 19, we show these metrics grouped by $Q(w)$ of the corresponding feature. Additionally, similarly to Section 5.1, we estimate the 95% CI for the $\sigma$ parameter of the distribution of $Q(w)$ when it is modeled as a log-normal distribution. We split features into two groups: those with `#code` above the median value and those below it. We perform the same split for `#punct`. The results, presented in Table 6, indicate that $Q(w)$ has heavier tails for the groups where `#code` (or `#punct`) is above the median, mirroring our results in Section 5.1.

### D.7   LOOK-AHEAD EXPERIMENT (SECTION 5.2)

We obtain a simple future token predictor by training a linear layer mapping from $r_{\theta,i}^k(x)$ to $x_{i+t+1}$ on a subset of the Pile dataset Gao et al. (2020):

$$\hat{x}_{i+t+1} = h_\theta^{L+1}(W_{\text{LA}} \cdot r_{\theta,i}^k(x))$$

Here $h_\theta^{L+1}$ is the frozen language modeling head of Gemma, and $t > 0$ is the distance of the look-ahead.

This approach resembles Linear Model Approximation of Pal et al. (2023), except we keep the language modeling objective instead of the reconstruction loss on the activations at the last layer.

After training, $W_{\text{LA}}$ defines a subspace in the space of the residual stream. Note that the weights of the SAE encoder define directions in the same space: indeed, one SAE feature can be seen is a linear projection of residual stream followed by an activation function. Thus, we can estimate how much a feature contributes to the look-ahead by checking how close the feature encoder vector lies to the subspace defined by $W_{\text{LA}}$.

Intuitively, if some linear direction $v$ in the residual stream of the model contains information crucial for predicting look-ahead linearly, $W_{\text{LA}}$ would learn to "read" from this direction, and columns proportional to $v$ would appear in $W_{\text{LA}}$.

To check that, we compute the angle $A(W_{LA}, w, r)$ between the feature direction $w$ and the subspace formed by $r$ main singular components of $W_{\text{LA}}$. Specifically, let $W_{\text{LA}} = U\Sigma V^T$ be the SVD decomposition of $W_{\text{LA}}$. Let $E$ be the matrix of SAE feature directions: the weights of the SAE encoder of the size $d \times n_{\text{features}}$. For each $r$ from 1 to $d$, we compute the projection matrix $P = V_r V_r^T$, feature projections $\hat{E} = PE$, and save the cosine distances between the corresponding columns in $E$ and $\hat{E}$.

In this way, for the given look-ahead distance $t$ and matrix $W_{\text{LA}}^t$, we obtain a matrix $C$ of size $d \times n_{\text{features}}$: $C_{rj}$ is the cosine similarity between $j$-th feature and its projection on $V_r V_r^T$. In Figures 7 and 20 we report the correlations between $Q(w)$ and $C_{rj}$ for each $r$.

## E FURTHER EXPERIMENTAL RESULTS

### E.1 TOY TASKS

Table 4: Accuracy of a trained model for the first token in the output phase. All trained models successfully solve the tasks, except for the myopic models trained on Conditioned Majority.

|  | non-myopic | | myopic | |
| --- | --- | --- | --- | --- |
|  | tied | 10-untied | tied | 10-untied |
| **Majority** | $1.00 \pm 0.00$ | $1.00 \pm 0.00$ | $1.00 \pm 0.00$ | $0.96 \pm 0.05$ |
| **Conditioned Majority** | $0.99 \pm 0.02$ | $1.00 \pm 0.00$ | $0.76 \pm 0.01$ | $0.76 \pm 0.09$ |

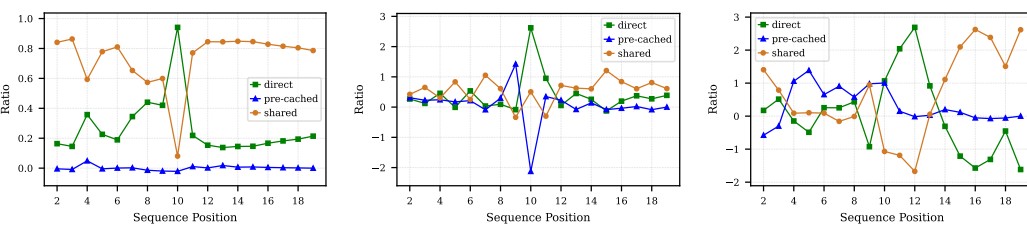

Figure 8: Normalized integrated influence components for the tasks of Majority **(left)** and Conditioned Majority (**center:** tied model, **right:** 10-untied model).

In models solving Majority, the features in the input phase seem to be predominantly shared, but that flips at the 10th token – exactly the point where the feature is needed to predict the immediate next token, which is the output for the task.

The images for Conditioned Majority may help explain the curious increase in the performance and probing accuracy of the 10-untied model compared to the tied model. We see that for the 10-untied model it is the pre-caching influence that causes the previous-token feature to be learned in the input phase, while for the tied model there is no clear pattern. This likely indicates that, instead of the robust circuit we expected, the tied model learned a less robust, possibly shortcut solution, due to the influence of shared gradients from the output phase.

### E.2 OTHELLO

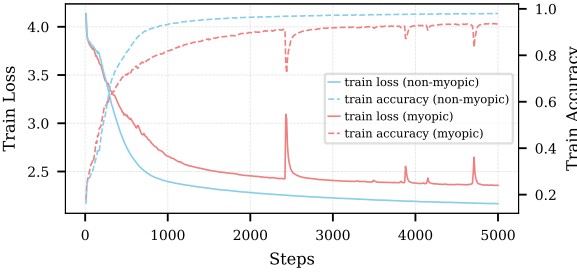

Figure 9: The training loss and accuracy curves of the myopic and non-myopic OthelloGPT models. The training of a myopic model is much less stable and it does not converge to the same performance as a non-myopic model.

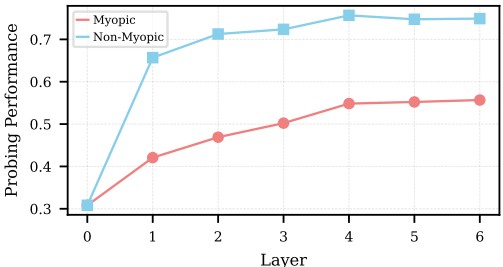

Figure 10: Performance of the probes trained to extract the board state from myopic and non-myopic Transformers for Othello. Myopic model shows a much lower degree of the board state representation.

The scores presented in this plot are lower than the ones reported in the prior literature (Nanda et al., 2023) due to the differences in evaluation methodology (we train the probes for regression and evaluate them using Pearson correlation, also unlike the prior work we include empty cells into evaluation). In addition, our model is smaller than the one used in the prior work (6 layers against 8 and a smaller hidden dimensionality), dictated by the limits of computational resources.

In Figures 11 and 12, we compare the direct and pre-cached influence components for different squares in the Othello experiment. Each plot is an average of the results for two close positions.

Interestingly, we find the results to be quite noisy and clearly non-stationary with respect to the position in the sequence. In general, it seems that the importance of the direct component generally goes down later in the game relative to the importance of the pre-cached component. However, we believe that more experiments are needed to make conclusive statements.

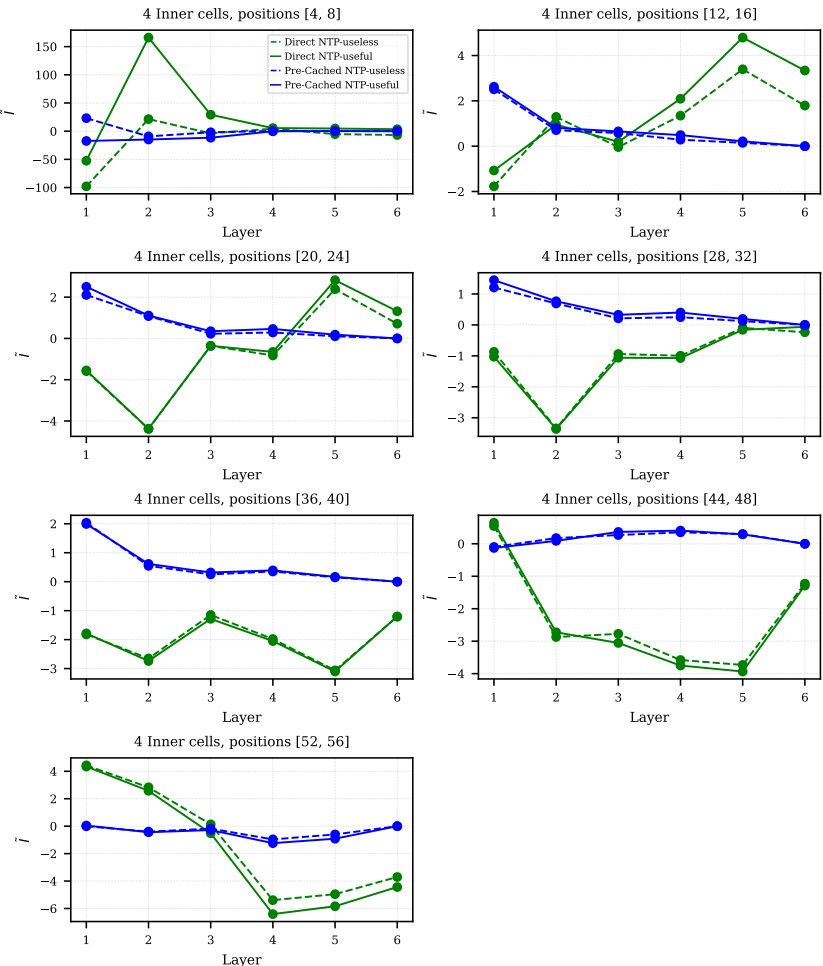

Figure 11: Integrated influence components at different positions for the 4 central cells in Othello. Each plot is an averaging of two positions.

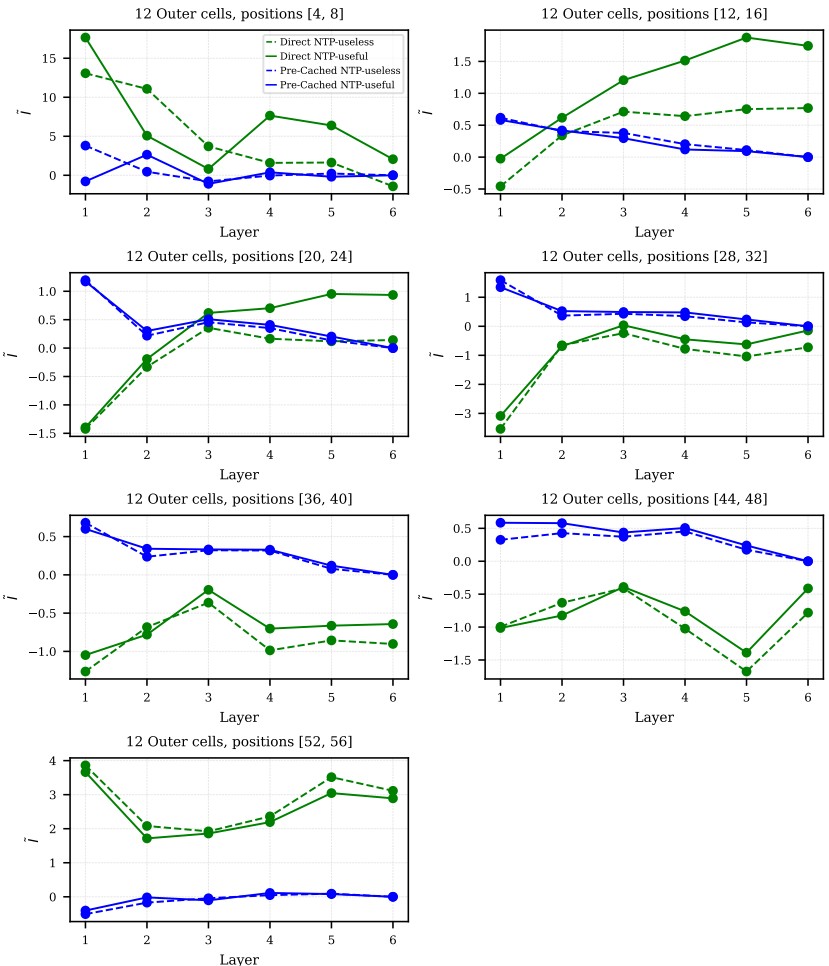

Figure 12: Integrated influence components at different positions for the 12 cells surrounding the center in Othello. Each plot is an averaging of two positions.

### E.3  SMALL LANGUAGE MODEL

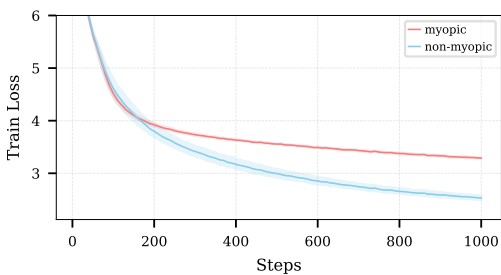

Figure 13: The training loss curves of the myopic and non-myopic small language models. It can be seen that a myopic LM has a consistently much higher loss from very early in training.

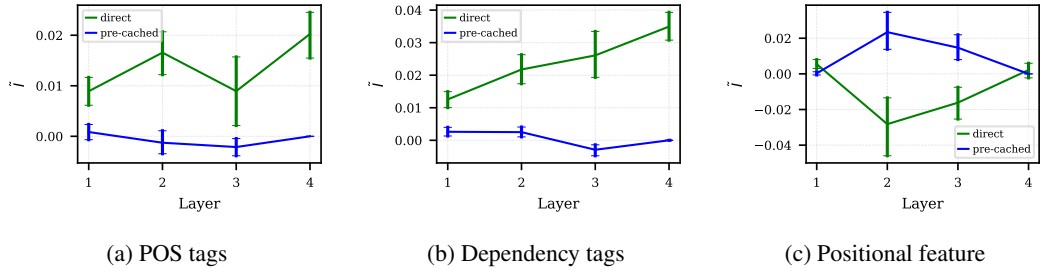

(a) POS tags        (b) Dependency tags        (c) Positional feature

Figure 14: Direct and pre-cached influence for different feature types (as in Figure 4) with 95% CI estimates for the average value at each layer.

| Feature category | Layer | $t$-statistic | $p_{\text{greater}}$ | $p_{\text{less}}$ |
|---|---|---|---|---|
| POS tags | 1 | 358640 | $5.8 \times 10^{-8}$ | 1.0 |
|  | 2 | 387970 | $3.1 \times 10^{-16}$ | 1.0 |
|  | 3 | 374375 | $5.5 \times 10^{-12}$ | 1.0 |
|  | 4 | 452676 | $3.3 \times 10^{-46}$ | 1.0 |
| Dependency tags | 1 | 269035 | $9.4 \times 10^{-18}$ | 1.0 |
|  | 2 | 297604 | $2.5 \times 10^{-34}$ | 1.0 |
|  | 3 | 301802 | $2.9 \times 10^{-37}$ | 1.0 |
|  | 4 | 350830 | $1.1 \times 10^{-80}$ | 1.0 |
| Positional | 1 | 4080 | $4.5 \times 10^{-8}$ | 1.0 |
|  | 2 | 1033 | 1.0 | $1.4 \times 10^{-7}$ |
|  | 3 | 1289 | 1.0 | $1.1 \times 10^{-5}$ |
|  | 4 | 3007 | $4.9 \times 10^{-2}$ | 0.95 |

Table 5: One-sided Wilcoxon test statistics comparing direct and pre-cached influence components, with p-values reported for each feature type and layer. In all layers, the direct influence for POS and dependency tags is significantly larger than the pre-cached influence, which is reversed in the middle layers for the positional feature.

### E.4  GEMMA 2

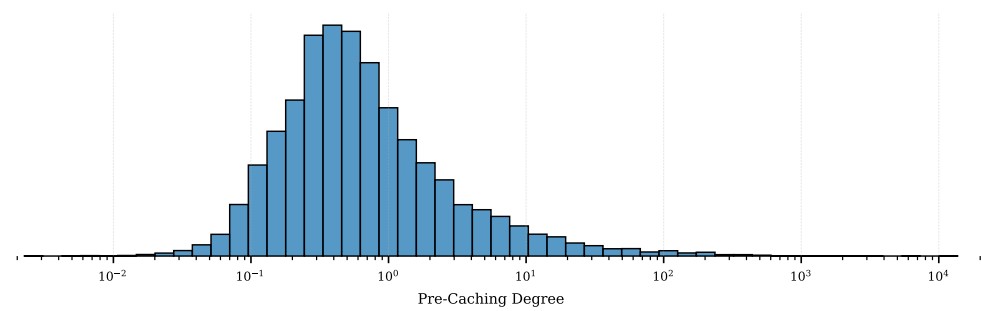

Figure 15: Distribution of $Q(w)$ (pre-caching degree) for the SAE features of Gemma 2 2B.

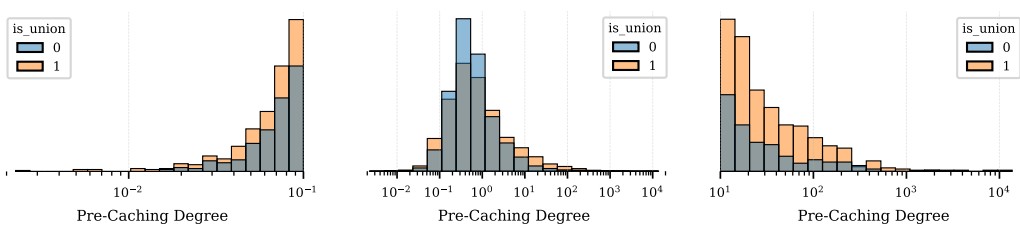

Figure 16: Distribution of $Q(w)$ (pre-caching degree) for the SAE features of Gemma 2 2B. The features classified as related to code, math, syntax, or text structure, are plotted orange; others are plotted blue. Images to the left and to the right show the tails of the distribution.

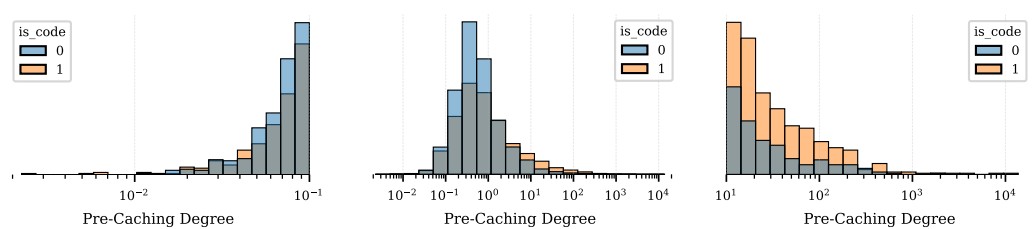

Figure 17: Distribution of $Q(w)$ (pre-caching degree) for the SAE features of Gemma 2 2B. The features classified as related to code or math are plotted orange; others are plotted blue. Images to the left and to the right show the tails of the distribution.

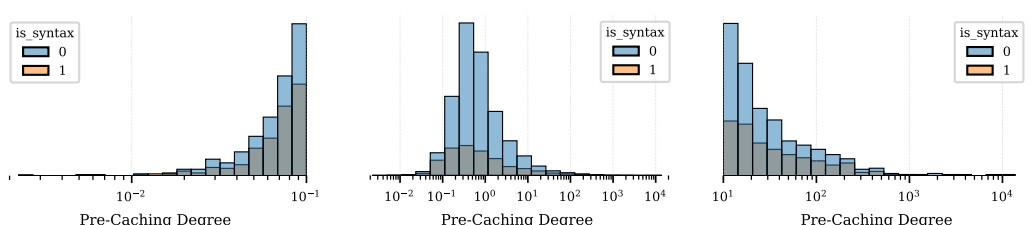

Figure 18: Distribution of $Q(w)$ (pre-caching degree) for the SAE features of Gemma 2 2B. The features classified as related to syntax or text structure, are plotted orange; others are plotted blue. Images to the left and to the right show the tails of the distribution.

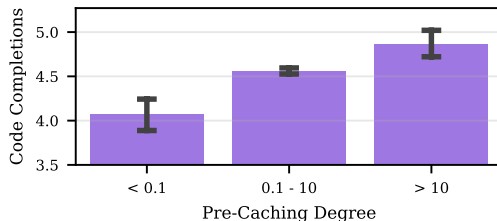 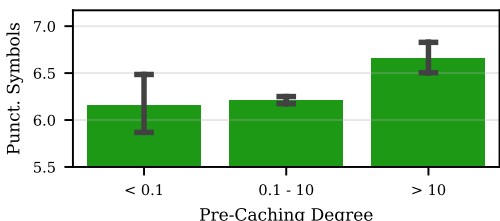

Figure 19: Number of unconditional generations classified as code (left) and average number of punctuation symbols (right) obtained by steering SAE features in Gemma 2 2B. Steering features with high pre-caching degree leads to both more code and more punctuation symbols being generated.

| Group | $\hat{\sigma}$ (95% CI) |
|---|---|
| #code above median | $1.479 \pm 0.030$ |
| #code below median | $1.428 \pm 0.020$ |
| #punct above median | $1.519 \pm 0.025$ |
| #punct below median | $1.351 \pm 0.022$ |

Table 6: Estimated scale parameter $\sigma$ of the log-normal distribution fitted to $Q(w)$ for different feature groups. For each group, we report the estimated $\hat{\sigma}$ and its 95% confidence interval. Features are split at the median value of #code and #punct.

| Feature | $Q(w)$ | Description |
|---|---|---|
| 15050 | 0.002 | HTML or CSS code referencing scripts and stylesheets |
| 3016 | 0.006 | programming constructs related to annotations and metadata in code |
| 3025 | 0.006 | French pronouns and their conjugations in various contexts |
| 4840 | 0.006 | aspects related to graphical user interface elements |
| 2127 | 0.007 | identifiers and numerical values in a structured format |
| 9124 | 0.010 | elements related to data structures and operations in programming contexts |
| 11737 | 0.011 | the article "An" used in various contexts |
| 4214 | 0.011 | numerical values or symbols, particularly those frequently associated with coding, data structures, or software libraries |
| 11655 | 0.013 | articles and determiners preceding nouns |
| 13070 | 0.014 | compiler directives and warning pragmas in code |

Table 7: 10 features with the lowest $Q(w)$ among the SAE features of Gemma 2 2B under study.

| Feature | $Q(w)$ | Description |
|---|---|---|
| 4592 | 13710.8 | programming-related terms and structures, particularly those associated with class and method definitions |
| 12285 | 6946.1 | references to web development, particularly relating to dependencies and library management |
| 6579 | 3308.5 | code structure and control flow statements |
| 15090 | 2675.1 | programming constructs and syntax elements |
| 10042 | 1853.5 | text related to software usage rights, permissions, and licensing terms |
| 13045 | 1641.9 | elements and objects that are part of a programming interface or user interface |
| 15829 | 1202.1 | function calls and their syntax within code |
| 6139 | 898.5 | technical references to Forms and related components in programming |
| 13552 | 871.2 | terms related to networking and software development |
| 2162 | 756.9 | colons or punctuation marks |

Table 8: 10 features with the highest $Q(w)$ among the SAE features of Gemma 2 2B under study.

| Feature | $Q(w)$ | Description |
|---|---|---|
| 9964 | 0.125 | terms related to standards and standardization processes |
| 14622 | 0.663 | terms related to poverty and marginalized communities |
| 5029 | 0.928 | details related to activities undertaken or actions witnessed in various contexts |
| 136 | 0.852 | terms associated with legal prohibitions and restrictions |
| 4010 | 0.435 | concepts related to logistics, regulations, and technical specifications |
| 13176 | 0.227 | references to events or gatherings |
| 2463 | 0.330 | references to assembly attributes in a programming context |
| 4685 | 0.109 | tokens related to identifiers or types in programming languages |
| 2803 | 0.171 | references to processes related to healthcare, particularly in the context of treatment and intervention strategies |
| 9598 | 0.560 | legal and political events or controversies |

Table 9: 10 random features with $Q(w)$ between 0.1 and 1.1.

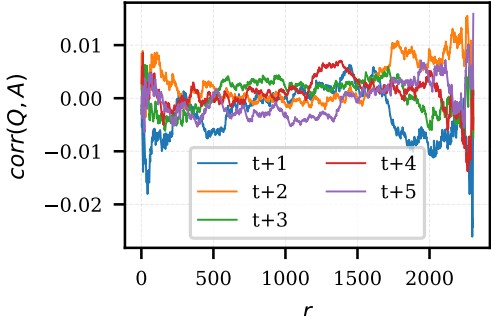

Figure 20: Pearson correlations between $\cos A(W_{LA}, w, r)$ and $Q(w)$ among the SAE features in Gemma 2.

STATEMENT ON LLM USAGE

We used LLMs to assist with text editing and with writing experimental code. We take full responsibility for all content in the paper and presented results.

