# OpenReview forum: "Understanding the Emergence of Seemingly Useless Features in Next-Token Predictors"
_ICLR.cc/2026/Conference — ICLR 2026 Poster_

### Official Review · Reviewer_XpYj · 2025-10-23

**Soundness:** 2
**Presentation:** 3
**Contribution:** 4
**Rating:** 6
**Confidence:** 3

**Summary:**

This paper studies why transformers trained on next token prediction (NTP) learn features that are not useful for predicting the immediate next token, such as abstract world models or complex syntactic structures. The authors propose that the standard "teleological" view (what features do in a final model) is insufficient. Instead, they adopt a "developmental" perspective, analyzing how features emerge from the gradient signal during training.

The paper's main contribution is a decomposition of the NTP gradient into three distinct components, based on the information flow relative to a specific hidden state at position $i$:
- direct learning, that is the standard gradient signal from the loss at position $i+1$. This component pushes toward features directly useful for predicting the next token.
- pre-caching, that we can see when the gradient signal from future losses (at positions $j > i+1$) flows back through attention to the state at position $i$. And this allows the model to learn to "pre-cache" information at position $i$ that will be needed for future predictions.
- circuit sharing: That is the gradient signal from losses at any position $j$ that updates the model's shared parameters ($\theta$) without passing through the specific state at position $i$. Because parameters are shared across all positions, a feature learned for a direct task at position $j$ can appear at position $i$ even if it's useless there.

The authors then introduce "myopic training" (to block pre-caching) and "m-untied training" (to block circuit sharing) to show that these mechanisms are necessary for learning NTP-useless features in toy tasks. And they propose a new method to quantify the "integrated influence" of each of the three gradient components on the development of a specific feature over the entire training process.

**Strengths:**

- The decomposition of the gradient signal into direct, pre-cached, and shared components is a powerful new way to think about feature emergence. It provides a principled answer to the fundamental question of why NTP models learn more than just $t+1$ prediction.
- The integrated influence attribution method (section 3.4) is computationally expensive, but it provides a concrete, quantitative tool for tracing the origin of a given feature back to its specific gradient source, moving beyond static, post-hoc analysis.

**Weaknesses:**

- The entire analysis pipeline (from probing to influence attribution (Def 3.2) to the $Q(w)$ proxy) depends on features being linearly represented in the residual stream. It's highly probable that many complex, abstract, or "useless" features are not linearly decodable, especially in middle layers. The paper's methodology doesn't take into account the emergence of such non-linear features.
- The paper's core premise relies on a binary split between NTP-useful and NTP-useless features. While for Othello this line is clear (does a square affect the next move?), with language it's way murkier. Is a Part-of-Speech tag NTP-useless? It might not be necessary for 90% of predictions, but it might be helpful for 10% (for example when the next token could be "jumped" or "jump").
- The the "integrated influence" method has a huge computational cost. It requires a full retraining of the model (or more, depending on the implementation) to analyze the development of features. This is acknowledged by the authors and is why they resort to a proxy for the LLM experiments.
- While the gradient paths for pre-caching and circuit sharing are defined as disjoint, their effects on parameter updates $\theta$ are deeply related. A pre-cache signal from a future loss at $j$ also contributes to the gradient on $\theta$, just as a shared-circuit signal from $j$ does. The attribution method for separating these influences (section 3.4, adapting Adam) is complex, and its robustness is not fully explored.
- The attribution method explains the origin of a known feature (defined by a probe direction $w$). It does not, by itself, discover features. The experiments rely on features found either via linear probes on ground-truth labels (toy tasks, Othello) or an unsupervised method (SAEs on Gemma). This creates a "chicken-and-egg" problem: to understand how a feature develops, you must already have a way to find it in the final model.
- The paper's entire analysis of the pre-trained LLM (Section 5) hinges on the $Q(w)$ metric being a faithful proxy for the developmental gradient influences. This link, established in proposition 5.1, is theoretically shaky. The proof provided in the appendix conflates a local, static analysis with a global, developmental history. It justifies the proxy by performing a first-order approximation (a Taylor expansion) around the final, trained model $\theta^*$, measuring the static causal effect of a feature after training is complete. It then equates this local, static measurement to the integrated influence (like, $I_{\text{pre-cached}}$) over the entire training run. This assumes, without evidence, that the gradient dynamics at the end of training are representative of the cumulative signals that created the feature from scratch, which is a very strong and unlikely leap.

**Questions:**

Your integrated influence method provides an interesting way to analyze the origin of a pre-defined feature (for example, from a probe or SAE). Could this developmental framework be inverted?
I mean, could you use the gradient components themselves during training as a discovery mechanism?

For example, could you monitor the training dynamics and look for directions in parameter space (or activation space) that consistently receive a high magnitude of "pre-cached" or "shared" gradient, but a low "direct" gradient? This might allow you to discover novel, abstract, or NTP-useless features as they form, rather than being limited to finding them post-hoc in the final model.

---

> ### Author Response · Authors · 2025-11-14
> **Response to Reviewer XpYj (1/2)**
>
> We thank the reviewer for their feedback and careful reading of the paper. Below, we address the issues pointed out by the reviewer.
>
>
> ## Responses to Concerns
> **1. Linear representation assumption**
>
> > The entire analysis pipeline ... depends on features being linearly represented in the residual stream. The paper's methodology doesn't take into account the emergence of non-linear features.
>
> We agree that our analysis is restricted to the linear features. However, there is substantial evidence that Transformers indeed tend to represent features linearly, including empirical evidence from small models, LLMs, and theoretical justifications (e.g., [1, 2, 3]). Moreover, the overwhelming majority of contemporary works in interpretability relies on this "linear representation hypothesis", with popular methods such as SAEs [4] or Logit/Tuned Lens [5] extracting only linear features. Some evidence that LLMs might represent features non-linearly exists [6], but remains extremely limited compared to the features found to be linear. Thus, we do not think that our focus on linear features is a substantial limitation.
>
> **2. NTP-useful vs NTP-useless features**
>
> > The paper's core premise relies on a binary split between NTP-useful and NTP-useless features...  Is a Part-of-Speech tag NTP-useless? It might not be necessary for 90% of predictions, but it might be helpful for 10% ...
>
> Under our definitions, such "mixed" features would still be considered NTP-useful (and "usefulness" of POS tags is shown by our results in Section 4.3). We agree that the line is much clearer for Othello and toy domains, which is exactly why we study them first.
>
> **3. Computational cost**
>
> > The the "integrated influence" method has a huge computational cost.
>
> We view our main contribution as the theoretical framework and the analysis of Transformers through its lens, not the method itself. As noted by the reviewer, we acknowledge the limitations of the attribution method in the paper.
>
> **4. The attribution method**
>
> > While the gradient paths for pre-caching and circuit sharing are defined as disjoint, their effects on parameter updates $\theta$ are deeply related. A pre-cache signal from a future loss at $j$ also contributes to the gradient on $\theta$, just as a shared-circuit signal from $j$ does.
>
> Our method accounts for this, and this is precisely the reason why we have to run separate forward passes with and without the stop-gradient operator applied to $r_{\theta, i}^k(x)$ (we explain it in Section 3.4 and Appendix B). The method for computing the components influence for a gradient update, thus, fully corresponds to the projections of the disjoint pre-cached and shared paths, as defined in Section 3.1, on the feature direction. The influences of all three components are, thus, separated, despite acting on the same weight vector $\theta$, which is the whole point of our attribution method.
>
> We hope this answer resolves the doubts of the reviewer. If they meant something else and we misinterpreted this comment, we ask the reviewer to clarify.
>
> > The attribution method for separating these influences (section 3.4, adapting Adam) is complex, and its robustness is not fully explored.
>
> We are not sure what the reviewer means by this comment. The adaptation of our method to Adam is equivalent to the standard Adam update rule while separating the contributions of each component to the update. We ask the reviewer to clarify their concern.
>
> **5. "Chicken-and-egg" problem**
>
> > The attribution method explains the origin of a known feature ... It does not, by itself, discover features ... This creates a "chicken-and-egg" problem ...
>
> We respectfully disagree with the reviewer that there is a "chicken-and-egg" problem. While computing influence components indeed requires training a linear probe first, the reverse is not true. Finding a feature via a linear probe is straightforward and does not require running the attribution first, which is why we are able to apply the method. Our overall pipeline is described in detail in Appendix B.
> Thus, we do not see any limitation here.
>
> **6. $Q(w)$ metric**
>
> > The paper's entire analysis of the pre-trained LLM (Section 5) hinges on the $Q(w)$ metric being a faithful proxy for the developmental gradient influences... It equates this local, static measurement to the integrated influence over the entire training run.
>
> We only claim that $Q(w)$ approximates the ratio of direct and pre-cached influence *around the weights of the final model*. Of course, equating $Q(w)$ to the influence integrated over the whole training run would be unreasonable, for example because a feature can potentially change its role during training. We will update the manuscript, making sure to state it more clearly.
>
> **Please see the second comment for the continuation of the response.**

---

> ### Author Response · Authors · 2025-11-14
> **Response to Reviewer XpYj (2/2)**
>
> ## Responses to Questions
>
> > Could this developmental framework be inverted? For example, could you monitor the training dynamics and look for directions in parameter space (or activation space) that consistently receive a high magnitude of "pre-cached" or "shared" gradient, but a low "direct" gradient?
>
> We haven't thought about this, but it is an interesting direction for future work. Potentially, as suggested by the reviewer, one could search for directions contained in the subspaces formed by pre-cached/shared but not direct updates. If there is a pre-cached/shared feature that we don't know about, it can potentially be found this way. We thank the reviewer for this suggestion and will consider working in this direction, as well as mention it in the section about future work.
>
> ## Conclusion
>
> We notice that the reviewer has given a low soundness score to the paper. We hope that our clarifications above address their doubts about it. If the reviewer is still concerned about the paper's soundness and has in mind potential flaws in our arguments, we are happy to engage with these thoughts as well, and ask the reviewer to communicate the remaining concerns to us.
>
> ## References
>
> [1] Park, Kiho, et al. "The Linear Representation Hypothesis and the Geometry of Large Language Models." ICML 2024.
>
> [2] Gurnee, Wes, and Max Tegmark. "Language Models Represent Space and Time." ICLR 2024.
>
> [3] Jiang, Yibo, et al. "On the Origins of Linear Representations in Large Language Models." ICML 2024.
>
> [4] Templeton, Adly, et al., "Scaling Monosemanticity: Extracting Interpretable Features from Claude 3 Sonnet", Transformer Circuits Thread, 2024.
>
> [5] Belrose, Nora, et al. "Eliciting latent predictions from transformers with the tuned lens." arXiv 2023.
>
> [6] Engels, Joshua, et al. "Not All Language Model Features Are One-Dimensionally Linear." ICLR 2025.

---

> ### Author Response · Authors · 2025-11-19
> **Response to Reviewer XpYj - Paper Updated**
>
> We have revised the paper incorporating your feedack. Please see our message to all reviewers for the summary of changes.
>
> In particular, regarding your feedback,
> * **weakness 6 ($Q(w)$ metric)** -> clarified that it approximates the ratio of influence components around the trained model and not along the whole training path.
> * **question 1 (inversing the method)** -> added this idea to the conclusion as a future work direction.
>
> We thank the reviewer for their feedback and hope that our responses and revisions address all of their concerns.

---

### Official Review · Reviewer_d3f7 · 2025-10-28

**Soundness:** 3
**Presentation:** 3
**Contribution:** 3
**Rating:** 6
**Confidence:** 3

**Summary:**

The paper is focused on the analysis of the gradient signal arising from the next token prediction task, which is ubiquitous in large language models and Transformers. This gradient, as already reported in the literature, does not only influence the features associated to the tokens which immediately precedes the last token, but spreads also to other tokens. The authors split the gradient signal in three components, direct (eq 1), pre-cashed (eq 2) and shared (eq 3), and they perform ablation and intervention experiments aimed at identifying the relative importance of these components in different transformers.  The analysis is performed on toy models, small transformers,  are also  on a large language model.

**Strengths:**

The analysis pipeline is well described, clear and neat, and might be potentially useful to other researchers. The ablation and interventional experiments are also well chosen and  well described.
Proposition 5.1 is interesting and also possibly useful to develop analysis pipelines.
The results on Gemma 2, and in particular fig 7, is interesting, as it provides evidence for a mechanism triggering directionality in LLMs.

**Weaknesses:**

It is not always clear what are the take-home messages.
The results in Fig3 show a performance  performance gap between NTP-useful and NTP-useless feature which has always the same sign, and gross monotonically with token index and layer index. This result seems pretty trivial.
In the analysis of TinyStory it is found that ablating precatching is detrimental for the training loss. This is also not surprising. A moderately unexpected result in that section is that syntax survives ablation.

**Questions:**

Why the residual stream of token i in an intermediate  layer k  (r_i^k) should have a special role in predicting the next token in the output layer (L+1). I understand that this must be the case for the penultimate layer, namely for r_i^L, but in the previous layers the token index is largely irrelevant, since the positional encoding is mixed and shuffled by the attention mechanism. Doesn't this make the decomposition arbitrary?

---

> ### Author Response · Authors · 2025-11-14
> **Response to Reviewer d3f7 (1/2)**
>
> We thank the reviewer for their feedback and positive assessment of our analysis pipeline.
>
> ## Responses to Concerns
>
> **Results in Fig3**
>
> > The results in Fig3 ... seem pretty trivial.
>
> We respectfully disagree with this assessment. The sole fact that Transformers trained for Othello tend to represent the cells that affect the next move better than the ones that don't was, for the first time, shown less than a year ago in [1]. Out of the multitude of other works on OthelloGPT (e.g., [2, 3, 4]), we are not aware of anyone else reporting this, which, in our view, shows that this fact is non so trivial to the research community.
>
> However, the result of [1] is purely empirical, whereas our contribution in Section 4.2 is to explain this phenomenon. This is achieved by computing the gradient influence using our approach, as reported in Figure 3 (right). We show that strength of the "model’s inductive bias toward next-token partitions of state", empirically found by [1], arises from the difference in signal via the gradient paths identified in Section 3, and show this with statistical significance. We also refine the result of [1] by observing that the signal for learning NTP-useless cells differs significantly from zero, meaning that the bias to learn those features exists as well, though it is less strong than the bias for representing the next-token partitions.
>
> If the reviewer finds these results aligning with expectations after reading the Section 3 of our paper that introduces the information flow decomposition framework, we consider this a sign that our framework introduced in Section 3 is useful, and that is the opposite of a limitation.
>
> We will make sure to discuss this more clearly in the updated version of the manuscript.
>
> **Analysis of TinyStories**
>
> > In the analysis of TinyStory it is found that ablating precatching is detrimental for the training loss. This is also not surprising. A moderately unexpected result in that section is that syntax survives ablation.
>
> Our main results in this section focus on the NTP-usefulness of the features encoded in the model. To our knowledge, this hasn't been done before. It is subjective whether these results are surprising, however, we believe that they are novel and relevant. A follow-up study can potentially investigate this in more detail, in particular focusing on more features and larger-scale runs.
>
> **Take-home messages**
>
> > It is not always clear what are the take-home messages.
>
> We hope that our comments above regarding Sections 4.2 and 4.3 bring more clarity. Speaking of a paper as a whole, we hope that a reader "takes home" from it a useful new perspective for thinking about the feature development in Transformers. As we show in Sections 4 and 5, this perspective can be useful to obtain novel findings and explain the prior ones.
>
> If the reviewer referred to other parts of the paper in this remark, we respectfully ask the reviewer to point them out, so that we can improve our messaging.
>
> ## Responses to Questions
>
> > Why the residual stream of token i in an intermediate layer k (r_i^k) should have a special role in predicting the next token in the output layer (L+1) ... in the previous layers the token index is largely irrelevant, since the positional encoding is mixed and shuffled by the attention mechanism.
>
> This is not entirely accurate. First, residual connections between attention layers build a direct pathway between $r_i^k$ and the logits for the $(i+1)$-th token, without "being shuffled by attention." Second, and more importantly, due to causal masking, the $i$-th token contains more information than any of the previous ones (since none of them can attend to the position $i$). For these two reasons, it is common in interpretability work to treat the hidden-layer representations of the $i$-th token as containing the features relevant for the corresponding position (e.g., [2, 5]). This approach is further corroborated by the recently discovered circuits in LLMs, where the correspondence between the position of the token and the features that it encodes holds [6].
>
> ## Conclusion
>
> Above, we have articulated the core contribution and take-home messages more explicitly. If the reviewer is still concerned about the paper's conclusions and motivation, we are happy to engage with these thoughts as well, and ask the reviewer to communicate the remaining concerns to us.
>
> ## References
>
> Please see the references in the next comment.

---

> > ### Comment · Reviewer_d3f7 · 2025-11-20
> >
> > I thank the authors for the clarifications. Concerning  the empirical tests, and in particular fig3,  I understand that the contribution is  explaining the observations following the approach in section 4.2.
> > Concerning the reply to my question on the residual stream, the argument made by the authors is not fully convincing. I would see some quantitative measure of how much information on token i in layer 1 is transferred to token i in layer k, and how much information is spread on other tokens. The assumption   seems at odds with the observation that intermediate layers encode semantic, which is largely insensitive to the position of the tokens.

---

> > > ### Author Response · Authors · 2025-11-24
> > >
> > > We thank the reviewer for engaging with our response.
> > >
> > > We believe there may not actually be any disagreement between us and the reviewer regarding their question about the residual stream.
> > >
> > > To make sure we do not misunderstand the reviewer, we summarize the conversation thread as we see it below.
> > >
> > > 1. In the paper, by introducing the information flow decomposition, we treat the role of tokens $t_i$ in predicting the token $t_{i+1}$ separately from the role of tokens $t_{j<i}$.
> > > 2. In the initial question, the reviewer asks if the decomposition is arbitrary, since the representations of all tokens $t_{\leq i}$ are mixed by attention.
> > > 3. In our response, we point out two features that make $t_i$ special compared to $t_{j<i}$: namely, residual connections linking $r_i^k$ to $\hat{t}\_{i+1}$ and the ability to attend to the whole subsequence $t_{1\dots i}$. These two features, which are true for $t_i$ but not for $t_{j<i}$, explain why it is justified to treat $t_i$ specially.
> > > 4. In their response, the reviewer states that the argument is not convincing, as some amount of information from the token $t_i$ leaks to other tokens and not only $t_{i+1}$, and that Transformers learn to encode position-insensitive semantic information.
> > >
> > > We do not in fact see a contradiction between (4) and our claims, in particular (3). In (3), we do not say that information doesn't spread between $t_{j < i}$ and $\hat{t}\_{i+1}$, but we only argue that $t_i$ has a special role, answering the question in (2). In fact, the pathways between $t_{j < i}$ and $\hat{t}\_{i+1}$ are exactly what we study in the paper as pre-caching. The quantitative measure of how much information is transferred from token $t_i$ to tokens other than $t_{i+1}$ is basically the pre-caching influence component that we at length discuss in the paper. Thus, we do not see what the reviewer disagrees with.
> > >
> > > To conclude,
> > > * *Does $t_i$ have a special role for predicting $t_{i+1}$?* -- Yes, because of (3).
> > > * *Does this mean that no information flows from $t_{j < i}$ to $t_{i+1}$?* -- No, and we study precisely that phenomenon in the paper.
> > >
> > > If we have misunderstood any of the arguments of the reviewer, we'd be glad if the reviewer can clarify. We hope that this brings more clarity and thank the reviewer for engaging in the discussion. We really appreciate the reviewer's time!

---

> ### Author Response · Authors · 2025-11-14
> **Response to Reviewer d3f7 (2/2)**
>
> ## References
>
> [1] Vafa, Keyon, et al. "What Has a Foundation Model Found? Using Inductive Bias to Probe for World Models." ICML 2025.
>
> [2] Li, Kenneth, et al. "Emergent world representations: Exploring a sequence model trained on a synthetic task." ICLR 2023.
>
> [3] Nanda, Neel, et al. "Emergent Linear Representations in World Models of Self-Supervised Sequence Models." BlackboxNLP Workshop 2023.
>
> [4] Yuan, Yifei, and Anders Søgaard. "Revisiting the othello world model hypothesis." ICLR 2025 Workshop on World Models.
>
> [5] Ghandeharioun, Asma, et al. "Patchscopes: A Unifying Framework for Inspecting Hidden Representations of Language Models." ICML 2024.
>
> [6] Lindsey, Jack, et al. "On the Biology of a Large Language Model." Transformer Circuits, 2025.

---

> ### Author Response · Authors · 2025-11-19
> **Response to Reviewer d3f7 - Paper Updated**
>
> We have revised the paper incorporating your feedack. Please see our message to all reviewers for the summary of changes.
>
> In particular, regarding your feedback,
> * **weakness 1 (take-home messages)** -> clarified the takeaways in Section 4.2.
>
> We thank the reviewer for their feedback and hope that our responses and revisions address all of their concerns.

---

### Official Review · Reviewer_PoFB · 2025-11-01

**Soundness:** 3
**Presentation:** 3
**Contribution:** 3
**Rating:** 6
**Confidence:** 3

**Summary:**

The authors propose studying the features developed during Next-Token prediction that are not immediately useful for the explicit next token prediction task. They do so by focusing on the residual stream and decomposing it in terms of the information signal: anticipatory(pre-cache), direct, and shared. The authors carefully decompose the loss into the three components, so that they may define feature mismatch and consider how this accumulates along a training trajectory. To ablate pre-caching, they borrow from Wu et al. the concept of myopic training. To ablate circuit sharing, they pick a layer, and use different, non-shared weights up to the layer (inclusive) and after the layer (exclusive). To estimate if the models (transformer-based) represent features linearly, they train position-specific linear probes to regress from the residual stream to the feature values. By ablating in the two ways described above, they find that on toy examples and GPT-2, features that do not immediately aid the next token prediction task do not emerge. Studying OthelloGPT, they confirm the large signal value for the direct component, congruent with an inductive bias to distinguish boards with similar valid next move sets; however, the indirect components persist, suggesting that NTP-useless features are still learnt. By ablating pre-caching, they also find that syntax performance does not degrade, while generation does. Shifting to large language models, the authors formulate an intervention study where they add a vector value to the residual stream and use it to estimate the direct vs pre-cached influence on a specific token. By looking at extreme values, they find that most SAE features relate to programming languages or formal properties of the input text; by considering a log-normal fit, the authors validate the hypothesis that pre-caching is required when emulating formal parsing.

**Strengths:**

- Studies the utility of emergent features that allow predictions beyond the next token in next token prediction pre-training
- Additional confirmation of hypothesis or previous literature results (Wu et al.'s pre-caching speculation/breadcrumbs hypothesis, OthelloGPT and board-state encoding)

**Weaknesses:**

- The premise of the study feels partially undefined: the assumption that transformers trained on NTP will converge to greedy decoding feels unintuitive/by fiat (this can be mitigated with a citation if appropriate when the problem space is defined in the Introduction)
- Very minor: the use of "rose" makes the information difficult to read (on a screen and borderline invisible on an eInk display)

**Questions:**

- Q1: A core assumption of this work seems to be (in my understanding) that an LM trained using NTP will focus on the greedy decode task, i.e. only the next token. This feels counterintuitive. Why is the greedy decode expected to be favoured against something closer to Viterbi? Is there an argument from expected algorithm complexity, and that we expect the neural net to favour lower complexity algorithms that approximate a correct hypothesis from the set of "valid" hypothesises, here sequence decoding algorithms?

---

> ### Author Response · Authors · 2025-11-14
> **Response to Reviewer PoFB**
>
> We thank the reviewer for their feedback.
>
> We note that the reviewer's summary of the paper contains some misunderstandings:
>
> > To ablate circuit sharing, they pick a layer, and use different, non-shared weights up to the layer (inclusive) and after the layer (exclusive).
>
> Our algorithm for m-untied training (ablating circuit sharing) splits the weights by token position, not layer index (Section 3.2).
>
> > By ablating in the two ways described above, they find that on toy examples and GPT-2, features that do not immediately aid the next token prediction task do not emerge.
>
> This is not our conclusion. We find NTP-useless features clearly emerging in toy tasks (Figure 2). In the GPT-2 experiment, we only conclude that syntactic features are not NTP-useless, not that such features fail to emerge entirely.
>
> ## Responses to Concerns/Questions
>
> We now discuss the issues pointed out by the reviewer.
>
> **1. Premise of the study**
>
> Regarding the reviewer’s concern about the premise of the study, here are some reasons why we do not think that the emergence of NTP-useless features is obvious a priori.
>
> * An arbitrary next-token predictor, unconstrained by the Transformer architecture, does not need to compute NTP-useless features (by our definition of NTP-uselessness). As an example, a Python program for Othello computing a function "sequence of moves" $\to$ "next legal move" doesn’t have to compute any NTP-useless features. Untied myopic Transformers, as shown in Section 4.1, do not learn NTP-useless features either.
> * Moreover, even with standard Transformer architecture, there is evidence in the literature that the gradient signal from NTP does not always lead to the development of NTP-useless features, jeopardizing performance on some tasks. [1] showed exactly the failure mode where Transformers converge to a greedy decoding instead of “planning ahead.” [2] observed a similar effect for more applied tasks such as natural language planning.
>
> We believe that the two arguments above show that the view of Transformers as “greedy” is, if not natural, certainly not counterintuitive, and that it shouldn’t be simply discarded without arguments. However, even though this is the starting point for our study, we do not view its refutation itself as our primary contribution. We believe that even if “the greedy hypothesis” is refuted, a principled way of analyzing the ways how Transformers learn to not be greedy deserves study on its own.
>
> Furthermore, in our work, we study not only the overall behavior of Transformers but also the "greediness"/pre-caching of specific features, which is relevant even if one already assumes that Transformers are not greedy.
>
> **2. Color scheme**
>
> > Very minor: the use of "rose" makes the information difficult to read (on a screen and borderline invisible on an eInk display)
>
> We thank the reviewer for noticing this. We will change the color scheme in subsequent versions of the manuscript.
>
> ## Conclusion
>
> We hope that the reviewer finds these additional details on our motivation convincing, and we are happy to engage in more discussion if the reviewer has further concerns. Given the lifted restriction on page limit for the post-rebuttal version of the manuscript, we will expand our discussion of the results in the paper.
>
> ## References
>
> [1] Bachmann, Gregor, and Vaishnavh Nagarajan. "The Pitfalls of Next-Token Prediction." ICML 2024
>
> [2] Thankaraj, Abitha, et al. "Looking beyond the next token." ICML 2025 Workshop on Long-Context Foundation Models.

---

> > ### Comment · Reviewer_PoFB · 2025-11-14
> >
> > Thank you for the response and clarifications regarding my misreading in the summary (a misread of Fig.2, most likely thinking of the layers indexing from the TinyStories instead of the token position of this section).
> >
> > I disagree that greedy vs non-greedy is non-obvious sans the [1] reference. Without conditioning or reducing expressive power, my prior would be to guess that a "caching" like mechanism develops. However, my main concern was either to ground this assumption in first principles or to reference a paper where the issue is observed. I think [1] serves just this role, so I feel my concern is sufficiently addressed.

---

> ### Author Response · Authors · 2025-11-19
> **Response to Reviewer PoFB - Paper Updated**
>
> We have revised the paper incorporating your feedack. Please see our message to all reviewers for the summary of changes.
>
> In particular, regarding your feedback,
> * **weakness 1 and question 1 (premise of the study)** -> cited papers reporting that Transformers are "greedy" in the introduction.
> * **weakness 2 (color scheme)** -> changed the color of shared component in the plots to orange.
>
> We thank the reviewer for their feedback and hope that our responses and revisions address all of their concerns.

---

### Official Review · Reviewer_zUNe · 2025-11-02

**Soundness:** 3
**Presentation:** 2
**Contribution:** 3
**Rating:** 6
**Confidence:** 3

**Summary:**

The paper proposes a theoretical framework that decomposes training gradients in next-token prediction into direct learning, pre-caching, and circuit sharing. The approach explains how models acquire features that do not contribute to immediate prediction. The framework is evaluated on toy tasks, OthelloGPT, TinyStories, and Gemma 2, showing consistent qualitative trends.

**Strengths:**

- **Originality.** The three-way gradient decomposition is novel and well formalized. It offers a developmental view of feature emergence that complements prior interpretability work.
- **Quality.** The theoretical analysis is rigorous and the experiments are well designed. Ablations on toy tasks and real models support the framework.
- **Structure and coherence.** The presentation follows a logical order from theory to empirical evidence. Figures and proofs are consistent with the stated claims.
- **Significance.** The analysis connects mechanistic interpretability and training dynamics, offering tools that could inform future studies.

**Weaknesses:**

- **Lack of empirical validation of \(Q(w)\).** Proposition 5.1 defines \(Q(w)\) as an influence proxy, but it is not validated against true influence ratios despite available data.
- **Modified optimizer.** The use of a non-standard Adam variant with separate moments for each gradient component could alter convergence. A control experiment with standard Adam is needed.
- **Correlation without causation.** The Gemma 2 analysis links high \(Q(w)\) to formal reasoning features but does not test causal impact on model behavior.
- **Limited statistical rigor.** Results are mainly qualitative plots without variance or confidence intervals. Basic statistical reporting would strengthen conclusions.
- **Accessibility.** The notation and level of abstraction make the paper difficult to follow for readers outside specialized training-dynamics research. More intuition or schematic explanations would improve clarity.
- **First-order assumption.** The framework relies on a linearized gradient view (small-step assumption), which may not capture non-linear effects in real LLM training.
- **Metric choice.** Definition 3.2 uses L2 distance for feature mismatch without justification. Other similarity measures could yield different insights.

**Questions:**

- **Feature filtering.** The paper excludes 1,819 of 16,384 Gemma 2 SAE features before analysis. What criteria were used, and could this exclusion bias the results?
- **Feature mismatch metric.** Why was L2 distance selected for Definition 3.2? Has cosine similarity or another metric been tested?

---

> ### Author Response · Authors · 2025-11-14
> **Response to Reviewer zUNe**
>
> We thank the reviewer for their feedback. Below, we address the issues pointed out by the reviewer:
>
> ## Responses to Concerns
>
> **1. Validation of $Q(w)$**
>
> > Proposition 5.1 defines (Q(w)) as an influence proxy, but it is not validated against true influence ratios despite available data.
>
> Unfortunately, computing true influence components for a large-scale LLM such as Gemma is not possible since it requires a lot of computational resources due to the necessity to retrain the model and perform multiple backpropagation calls for each training step. This is the reason why we resort to $Q(w)$ in the first place. We ask the reviewer to clarify what available data they are referring to in this comment.
>
> **2. Modified optimizer**
>
> > The use of a non-standard Adam variant with separate moments for each gradient component could alter convergence. A control experiment with standard Adam is needed.
>
> We would like to point out that we use standard Adam without any modifications. Keeping separate moments for each gradient component is a reformulation that is mathematically equivalent to the standard update rule of Adam. Thus, our optimizer is standard and no control experiment is needed.
>
> **3. Causal analysis for Gemma 2**
>
> >  The Gemma 2 analysis links high (Q(w)) to formal reasoning features but does not test causal impact on model behavior.
>
> We agree that a causal experiment would be useful to test the robustness of the link between $Q(w)$ and the feature semantics. We will run an experiment involving SAE steering and report the results during the discussion phase.
>
> **4. Statistical rigor**
>
> > Results are mainly qualitative plots without variance or confidence intervals. Basic statistical reporting would strengthen conclusions.
>
> We respectfully disagree with this assessment of our results. We report confidence intervals to support our statements where appropriate, including Sections 4.2, 4.3, and 5.1. As noted in the discussion with Reviewer PgxN, during the discussion phase we will run an additional statistical test for Section 4.3. We ask the reviewer to clarify where else they see a need for statistical reporting. Without specifics, it is hard for us to make necessary actions to address this comment.
>
> **5. Accessibility**
>
> > The notation and level of abstraction make the paper difficult to follow for readers outside specialized training-dynamics research. More intuition or schematic explanations would improve clarity.
>
> We ask the reviewer to point out which specific parts would benefit from clarification, so that we can implement this suggestion.
>
> **6. First-order assumption**
>
> > First-order assumption ... which may not capture non-linear effects in real LLM training.
>
> This assumption, implied by analyzing loss gradients while ignoring higher-order derivatives, is common in prior work (e.g., [1, 2]). Moreover, this is implicitly assumed by training with Adam in general, since Adam is a first-order optimizer.
>
> **7. Metric choice**
>
> > Metric choice. Definition 3.2 uses L2 distance for feature mismatch without justification. Other similarity measures could yield different insights.
>
> We use L2 because it is the standard metric in regression tasks. Further explanation is provided in the section below.
>
> ## Responses to Questions
>
> > Feature filtering. The paper excludes 1,819 of 16,384 Gemma 2 SAE features before analysis. What criteria were used, and could this exclusion bias the results?
>
> We discard the features with low influence on all generated tokens. The procedure is explained in detail in Appendix D.4. Including those features would lead to noisy results since they would not provide informative signal.
>
> > Feature mismatch metric. Why was L2 distance selected for Definition 3.2? Has cosine similarity or another metric been tested?
>
> As mentioned above, we use L2 since it is the standard metric in regression tasks. We do not think that cosine similarity is a good metric for our purposes, as it can assign high similarity to distinct predictions (if their magnitude is different).
>
> ## Conclusion
>
> We will publish the statistical tests for the influence components in Section 4.3 and the results of the steering experiment in Section 5.1 as a separate comment here as soon as they are ready. We hope that our comments address all concerns of the reviewer.
>
> ## References
>
> [1] Lan, Janice, et al. "Lca: Loss change allocation for neural network training." NeurIPS 2019.
>
> [2] Mircea, Andrei, et al. "Training Dynamics Underlying Language Model Scaling Laws: Loss Deceleration and Zero-Sum Learning." ACL 2025.

---

> ### Author Response · Authors · 2025-11-19
> **Response to Reviewer zUNe - Paper Updated**
>
> We have revised the paper incorporating your feedack. Please see our message to all reviewers for the summary of changes.
>
> In particular, regarding your feedback,
> * **weakness 3 (causal analysis for Gemma 2)** -> added a steering experiment.
> * **weakness 4 (statistical rigor)** -> added analysis of statistical significance for the results in Section 4.3. It supports our findings from the original version of the paper.
>
> We thank the reviewer for their feedback and hope that our responses and revisions address all of their concerns.

---

### Official Review · Reviewer_PgxN · 2025-11-02

**Soundness:** 3
**Presentation:** 3
**Contribution:** 3
**Rating:** 4
**Confidence:** 3

**Summary:**

The paper studies the formation of "useless" next-token prediction features (NTP-useless) in Transformers. The authors decompose the loss gradient signal according to its relationship with the residual stream activations. The gradient signal is divided into direct (containing information from the next-token prediction), pre-cached (containing information from future tokens), and shared components (gradient updates to parameters reused across positions that do not depend on the current residual stream activation). They then propose a measure of the approximated influence of each of these signals on the emergence of a feature in the final model. To validate this, they train ablated models under two modified regimes: myopic training (blocking pre-caching gradients) and m-untied training (preventing feature sharing across positions). In toy tasks (Majority voting and Conditioned Majority) they observe that ablated two-layer Transformers fail to learn the NTP-useless required to solve the task. In OthelloGPT, board states are found to emerge indirectly through pre-caching and circuit sharing. In small GPT2-like language models trained on TinyStories, pre-caching improves overall language modeling performance, while syntactic features are learned primarily through direct gradients. Finally, in Gemma 2, features with stronger apparent pre-caching influence tend to be associated with formal or code-like structures.

**Strengths:**

- The paper introduces a novel decomposition of gradient signal into direct, pre-cached and shared components. This framework extends [Wu et al., 2024](https://arxiv.org/pdf/2404.00859)'s work by studying the emergence of features as a function of the different gradient signals.
- The authors offer a valuable methodological advance for interpreting training dynamics.
- The framework is used in four diverse settings: toy algorithmic tasks, OthelloGPT, TinyStories, and Gemma 2 2B, and is used to extract novel insights into how different sources of gradient signal shape feature emergence.

**Weaknesses:**

- The influence estimation framework requires retraining the models with the same random seed and data order, which is infeasible for large-scale models. This limits the method’s practical applicability.
- The role of "pre-cached" features in text generation models (section 4.3) is unclear. No statistical significant results showed. Absolute levels of influence are not informative.
- In the pre-cached features analysis on Gemma 2 (section 5.1), the authors claim that SAE features with high pre-cache influence are related to "programming or formal structure of the input text". However, these type of features are also found on the opposite end (low pre-cache influence, Figure 6). Overall, this finding is based on qualitative examples, with no statistical validation.
- Claimed support for the breadcrumbs hypothesis of [Wu et al., 2024](https://arxiv.org/pdf/2404.00859) (look-ahead behavior in LLMs arises not from explicit planning but from the overlap between the features required to predict tokens at different positions) is based on a single correlational experiment.

**Questions:**

- Does steering with the high $Q(w)$ SAE features have an influence in the generated output?
- Where does $y$ come from in the _influence_ definition (Definition 3.3).

---

> ### Author Response · Authors · 2025-11-14
> **Response to Reviewer PgxN**
>
> We thank the reviewer for acknowledging our contributions and providing thoughtful feedback and comments. Below, we address the issues pointed out by the reviewer.
>
> ## Responses to Concerns
>
> **1. Influence estimation framework**
>
> > The influence estimation framework requires retraining the models ... This limits the method’s practical applicability.
>
> We view our main contribution as the theoretical framework and the analysis of Transformers through its lens, not the method itself. We do not expect our method for computing the integrated influence to be applied for large-scale models, and we explicitly acknowledge this in our conclusion. We will update the manuscript, making sure to state this more clearly.
>
> **2. Results in Section 4.3**
>
> > The role of "pre-cached" features in text generation models (section 4.3) is unclear. No statistical significant results showed. Absolute levels of influence are not informative.
>
> We do report confidence intervals for the loss achieved by myopic and non-myopic models (line 360), which are separated from each other, clearly showing that pre-caching helps language modeling. Regarding our further results in this section (comparing the influence components of different feature types), we agree that statistical tests could be useful to corroborate the results. We will update the manuscript and include them.
>
> **3. Pre-cached features in Gemma 2**
>
> > In the pre-cached features analysis on Gemma 2 (section 5.1), the authors claim that SAE features with high pre-cache influence are related to "programming or formal structure of the input text". However, these type of features are also found on the opposite end (low pre-cache influence, Figure 6).
>
> Indeed, in Section 5.1 we find that a higher proportion of "formal reasoning" features is exhibited by both tails of the distribution of $Q(w)$. This does not contradict our claims, but we agree that in some of our statements we may accidentally bias the reader toward misreading the result. We thank the reviewer for noticing this. We will adjust our phrasing in the paper.
>
> > Overall, this finding is based on qualitative examples, with no statistical validation.
>
> We respectfully disagree with this assessment. In Section 5.1 (line 423) we report the confidence intervals for the variance of $Q(w)$ across formal and non-formal features and find that the variance is substantially higher for formal features. This means that they are less concentrated around the mode and display heavier tails with extreme values of $Q(w)$, which is exactly the result we claim in this section. We would appreciate other suggestions from the reviewer on the appropriate ways to test this hypothesis.
>
> **4. Breadcrumbs hypothesis**
>
> > Claimed support for the breadcrumbs hypothesis of Wu et al., 2024 is based on a single correlational experiment.
>
> Testing the hypothesis of Wu et al. is not our primary goal, which is why we did not include more experiments on it. We do believe, however, that our result indeed represents the evidence supportive of the breadcrumbs hypothesis. If it were wrong (and the pre-caching hypothesis of Wu et al. were right), the features with stronger pre-caching influence would also contribute more to look-ahead; we observe the opposite.
>
> ## Responses to Questions
>
> > Does steering with the high $Q(w)$ SAE features have an influence in the generated output?
>
> We agree that a steering experiment would be useful to test the robustness of the link between $Q(w)$ and the feature semantics. We will run such experiments and report the results during the discussion phase.
>
> > Where does $y$ come from in the influence definition (Definition 3.3).
>
> This is a typo, $y$ shouldn't be there. Thank you for noticing it.
>
> ## Conclusion
>
> We will publish the statistical tests for the influence components in Section 4.3 and the results of the steering experiment in Section 5.1 as a separate comment here as soon as they are ready. We hope that our comments address all concerns of the reviewer.

---

> ### Author Response · Authors · 2025-11-19
> **Response to Reviewer PgxN - Paper Updated**
>
> We have revised the paper incorporating your feedack. Please see our message to all reviewers for the summary of changes.
>
> In particular, regarding your feedback,
> * **weakness 1 (influence estimation framework)** -> stated its limitations more clearly in the paper.
> * **weakness 2 (results in Section 4.3)** -> added analysis of statistical significance for the results in this section. It supports our findings from the original version of the paper.
> * **weakness 3 (pre-cached features in Gemma 2)** -> clarified the presentation of our results in the abstract.
> * **question 1 (steering)** -> added a steering experiment. The answer to the question posed by the reviewer (Does steering with the high $Q(w)$ SAE features have an influence in the generated output?) is positive.
> * **question 2 (influence definition)** -> fixed the typo.
>
> We thank the reviewer for their feedback and hope that our responses and revisions address all of their concerns.

---

### Author Response · Authors · 2025-11-19
**Paper Updated**

Thanks to all reviewers for their feedback and their time! We really appreciate it.
We have updated the paper taking into consideration the reviewers' feedback. We ask the reviewers to check the revised version. The parts that we added or updated compared to the original version are highlighted in blue.

Specifically, the following changes were implemented:

1. Added an experiment testing the causal link between high $Q(w)$ and formal reasoning in the features of Gemma 2 by steering these features and inspecting the changes in generated text. We observed that steering features with high $Q(w)$ results in more code and more punctuation being generated (however, this is not observed for features with low $Q(w)$) *[suggested by reviewers PgxN and zUNe]*.
2. Demonstrated statistical significance of the results in Section 4.3 *[suggested by reviewers PgxN and zUNe]*.
3. Clarified the takeaways in Section 4.2 *[suggested by reviewer d3f7]*.
4. Cited papers reporting that Transformers are "greedy" in the introduction *[suggested by reviewer PoFB]*.
5. Stated more clearly that the attribution method is not applicable to large models *[suggested by reviewer PgxN]*.
6. Clarified our results with Gemma 2 in the abstract *[suggested by reviewer PgxN]*.
7. Clarified that $Q(w)$ approximates the ratio of influence components around the trained model and not along the whole training path *[suggested by reviewer XpYj]*.
8. Added a paragraph about future work *[suggested by reviewer XpYj]*.
9. Changed the color for the shared component in the plots *[suggested by reviewer PoFB]*.
10. Fixed the typo with $y$ *[suggested by reviewer PgxN]*.

With these updates, we believe that we address all concerns and suggestions of the reviewers; we thank them again for their feedback. If the reviewers have more actionable suggestions or concerns, we are happy to implement those as well.

---

### Author Response · Authors · 2025-12-01
**Summary of the discussion**

We are writing this update shortly after the decision to stop author-reviewer discussion and reassign Area Chairs was announced, as a summary intended to help the new Area Chair metareview the paper. It would be obviously untrue to pretend that we do not have our own stake in the outcome, but when assembling this summary we tried to stay as objective and faithful to the original reviews as possible.

Below, we discuss separately the contribution, soundness, and presentation of our paper, together with the reviewers' comments on each.

## Contribution

All reviewers evaluated our contribution as good or excellent (reviewer XpYj). Most reviewers explicitly acknowledge the novelty and insight of our proposed gradient decomposition framework (PgxN, zUNe, XpYj) or mention the contribution of our analysis pipeline among the strengths of the paper (PgxN, d3f7, XpYj).

Some concerns were raised regarding the paper's contributions:

* Reviewer PoFB noted that "the premise of the study feels partially undefined." We addressed that weakness by adding prior work motivating our research questions, and in their response to our rebuttal the reviewer acknowledged that their concern was sufficiently addressed.
* Reviewer d3f7 remarked on the lack of clarity in some of the take-home messages, stating that some of our results are not surprising. In our response, we clarified the novelty of our results. The reviewer thanked us for the clarification and mentioned that they understand the contribution, but did not say whether the clarifications changed their opinion.
* Some reviewers (PgxN, XpYj) mention the computational cost of our analysis pipeline among the weaknesses. We acknowledge this but do not view it as a major weakness, since the method is a secondary contribution compared to the framework itself.

## Soundness

All reviewers rated soundness as good, except reviewer XpYj, who rated it as fair. Some reviewers explicitly mentioned the coherence of our theoretical and empirical analysis among the strengths of the paper (zUNe, d3f7).

The main concerns about soundness are:

* Reviewers PgxN and zUNe commented on the limited statistical analysis of our results, specifically in Section 4.3. In the revised version, we added more statistical analysis in that section. We believe that we demonstrate statistical significance everywhere such analysis is natural (Sections 4.2, 4.3, and 5.1).
* Reviewers PgxN and zUNe also pointed out the lack of a causal link between estimated pre-caching influence and feature semantics in Section 4.1. To address this, we added an intervention experiment suggested by reviewer PgxN. The new results aligned with our initial findings.
* In our view, the lower soundness score from reviewer XpYj stems from several misunderstandings of our experimental design and conclusions. Specifically, we disagree with the reviewer on the supposed "chicken-and-egg" issue, unrobustness of the influence computation method, and the logical leap in Proposition 5.1. We posted a clarification message within three days of receiving the reviews and incorporated the clarifications into the revised manuscript, but the reviewer did not respond, so we do not know whether they would have changed their mind.

Reviewers zUNe and XpYj also raised concerns about some of our assumptions which, as we clarified in our responses, are standard in prior work. We do not discuss these points here to keep the summary concise and refer the Area Chair to our discussion with those reviewers.

## Presentation

All reviewers rated presentation as good, except reviewer zUNe, who rated it as fair. However, the same reviewer also listed "structure and coherence" as a strength. They noted that "the notation and level of abstraction make the paper difficult to follow for readers outside specialized training-dynamics research," but did not mention any concrete parts needing improvement and did not respond to our request for clarification. No other reviewers raised concerns about the presentation.

## Overall

All reviewers except reviewer PgxN gave a positive overall rating. We believe that we addressed all their concerns, so their post-rebuttal scores could reasonably have been even higher.

Reviewer PgxN flagged four weaknesses, two of which do not affect the paper's main contribution (the gradient decomposition framework), and two of which we addressed directly by adding more analysis. For that reason, we believe the reviewer might have increased the score had the discussion not been cut short by the leak.

---

### Meta-Review · Area_Chair_DBKX · 2026-01-06

**Summary:**

This paper proposes a theoretical framework to decompose the next-token prediction gradient into direct, pre-cached, and shared components, explaining how "seemingly useless" features (like world models) emerge during training. The decision to accept is informed by the authors' successful rebuttal which resolved primary concerns regarding statistical rigor and causal evidence, leaving only minor limitations regarding the method's computational cost and linear assumptions which do not outweigh the novelty of the gradient decomposition framework.

**Reviewer Concerns:**

During the rebutttal, the authors have reasonably addressed major conerns about the statistical significance of the results, the causal link between pre-caching influence and feature semantics (via a new intervention experiment), and the clarity of the study's premise/motivation. some minor concerns i.e., the computational cost of the full influence estimation pipeline, the reliance on linear feature assumptions, and the complexity of the notation may still remain but not a main blocker for accepting this paper as the theoretical framework provides a valuable contribution to understanding training dynamics.

**Reviewer Scores:**

Reviewer PgxN (Score: 4): This reviewer would likely have increased the score (to 6). as the authors directly addressed their two main actionable weaknesses (adding statistical validation and a causal intervention experiment) and provided arguments for why the other concerns (computational cost) did not diminish the primary theoretical contribution.

---

### Decision · Program_Chairs · 2026-01-26

Accept (Poster)